# Unimodal statistical learning produces multimodal object-like representations

Gábor Lengyel[1]*, Goda Žalalytė[2], Alexandros Pantelides[2], James N Ingram[2,3], József Fiser[1]†*, Máté Lengyel[1,2]†*, Daniel M Wolpert[2,3]†*

[1]Department of Cognitive Science, Central European University, Budapest, Hungary; [2]Computational and Biological Learning Lab, Department of Engineering, University of Cambridge, Cambridge, United Kingdom; [3]Zuckerman Mind Brain Behavior Institute, Department of Neuroscience, Columbia University, New York, United States

**Abstract** The concept of objects is fundamental to cognition and is defined by a consistent set of sensory properties and physical affordances. Although it is unknown how the abstract concept of an object emerges, most accounts assume that visual or haptic boundaries are crucial in this process. Here, we tested an alternative hypothesis that boundaries are not essential but simply reflect a more fundamental principle: consistent visual or haptic statistical properties. Using a novel visuo-haptic statistical learning paradigm, we familiarised participants with objects defined solely by across-scene statistics provided either visually or through physical interactions. We then tested them on both a visual familiarity and a haptic pulling task, thus measuring both within-modality learning and across-modality generalisation. Participants showed strong within-modality learning and 'zero-shot' across-modality generalisation which were highly correlated. Our results demonstrate that humans can segment scenes into objects, without any explicit boundary cues, using purely statistical information.
DOI: https://doi.org/10.7554/eLife.43942.001

**\*For correspondence:**
lengyel.gaabor@gmail.com (GL);
fiserj@ceu.edu (JF);
m.lengyel@eng.cam.ac.uk (ML);
wolpert@columbia.edu (DMW)

†These authors contributed equally to this work

**Competing interests:** The authors declare that no competing interests exist.

## Introduction

The coherent organization of information across different modalities is crucial for efficiently interacting with the world and lies at the heart of the concept of what defines an object (*Amedi et al., 2001*; *Pascual-Leone and Hamilton, 2001*; *Streri and Spelke, 1988*). Classic theories place visual boundaries, edges and contrast transitions, at the very core of the process by which we segment objects from the environment (*Marr, 1982*; *Peterson, 1994*; *Riesenhuber and Poggio, 1999*; *Spelke, 1990*; *von der Heydt et al., 1984*; *Zhou et al., 2000*). However, such boundary cues are insufficient for successful segmentation alone as they can both lead to false object boundaries within objects (e.g. the stripes of a zebra) and miss boundaries between objects (e.g. the illusory contours of the Kanizsa triangle). Here we propose that it is not boundaries per se that are required for object segmentation but instead they are just one example of a more general principle that leads to object representations: consistent statistical properties — be they visual or haptic.

Statistical learning mechanisms, sensitive to the co-occurrence statistics of individual constituent elements (syllables, shapes, etc), have been demonstrated to be capable of segmenting inputs even when low-level segmentation cues were entirely uninformative (*Aslin et al., 1998*; *Fiser and Aslin, 2005*; *Orbán et al., 2008*; *Saffran et al., 1996*) but it is unknown whether the representations they create go beyond forming abstract 'units' (formalised as latent causes; *Gershman and Niv, 2010*; *Orbán et al., 2008*) or lead to real object-like representations. To explore this, we used a set of artificial stimuli, in which the statistical contingencies defining objects had, by design, no correlation with boundary cues. This avoided the problem that, under natural conditions, boundary cues and

edges can be correlated with the statistical contingencies of objects (*Geisler et al., 2001*). As a minimal requirement for what an object is — 'a material thing that can be seen and touched' (*Oxford Dictionaries, 2017*) — we reasoned that the litmus test of whether the representations that emerge during statistical learning were really object-like was to measure 'zero-shot' generalisation (*Fu et al., 2014*; *Lampert et al., 2009*). That is, participants should be able to predict haptic properties of objects based on just visual statistics, without any prior haptic experience with them and without receiving any form of feedback in the generalisation task, and vica versa, they should be able to predict visual properties based on haptic statistical exposure alone.

## Results

We created an inventory of artificial 'objects', such that each object was defined as a unique *pair* of unfamiliar shapes (*Figure 1A*, inventory, colouring and gaps within pseudo pairs for illustration only). Note that only the individual shapes, but not the pairs defining the objects of the inventory, had visible boundaries. Therefore, boundary cues were uninformative with regard to the object identities, and instead participants could only rely on the statistical contingencies among the shapes that we created in either the visual or the haptic modality during an exposure phase. We then examined how the information extracted from the visual or haptic statistics affected performance on both a visual familiarity and a haptic pulling test, thus measuring within-modality learning as well as across-modality generalisation of statistical information.

We first examined visual learning and visual-to-haptic generalisation. During exposure, participants (N = 20, after exclusion, see Materials and methods) experienced a sequence of visual scenes, each consisting of a spatially contiguous cluster of 6 shapes displayed on a grey square (*Figure 1A*, exposure, top). Unknown to the participants, each 6-element scene was constructed by combining three of the objects from the inventory of true pairs (coloured explanatory diagrams shown above displays). Therefore, the objects could only be identified based on the consistent visual co-occurrence of their constituent shapes across scenes as participants did not have any experience with the scenes' haptic properties. In a sequence of trials, we then tested participants' visual familiarity by requiring them to choose which of two pairs in a trial was more familiar: a true pair or a 'chimeric' pseudo pair constructed of two shapes belonging to two different true pairs of the inventory (*Figure 1A*, test, top). This test is analogous to comparing familiar scenes that contain real-world objects (e.g. rabbits or deers), and thus comply with the known statistical regularities of the world, with unfamiliar scenes containing chimaeras (e.g. a wolpertinger — a mythical hybrid animal with the head of a rabbit, the body of a squirrel, the antlers of a deer, and the wings of a pheasant; *Wikipedia contributors, 2018*). In line with previous results (*Fiser and Aslin, 2005*; *Fiser and Aslin, 2001*), we found that mere visual observation of the exposure scenes enabled participants to perform significantly above chance in the visual familiarity test (*Figure 2A*, black dots: visual familiarity performance for individuals quantified by fraction correct, green dot and error bar: group average 0.77 [95% CI: 0.67–0.87], $t(19)$ = 5.66, p=$1.9 \cdot 10^{-5}$, Bayes factor = 1253). That is, in novel test scenes participants judged 'true pairs' more familiar than 'pseudo pairs', despite having seen all constituent shapes an equal number of times.

Critically, the exposure to visual statistical contingencies also generalised to participants' judgements as to the force required to pull apart novel compound objects. In order to provide participants with general experience about the forces associated with pulling objects apart in different configurations in our set-up, but without any reference to the objects of the shape-inventory, we pre-trained them on a task that required them to pull apart scenes consisting of coloured rectangles as objects which thus had clear boundaries (*Figure 1—figure supplement 1*). Participants then performed the main pulling task which used the shapes of the inventory, such that each scene consisted of two true pairs of the inventory, arranged as a 2×2 square (*Figure 1A*, haptic pulling test). On each trial, participants had to pull on a scene in a predetermined direction with the minimal force they thought was necessary to separate the scene (into two vertical pieces for horizontal pulling and vice versa; *Figure 1B*, right). Crucially, we simulated clamps at the corners of the scene that prevented it from actually separating, so that participants received no haptic or visual feedback as to whether they exerted the correct amount of force, and thus their performance must have been solely based on what they had learned about the visual statistics of the objects during the exposure phase. Specifically, given the pre-training, and their knowledge of the objects of the inventory, participants

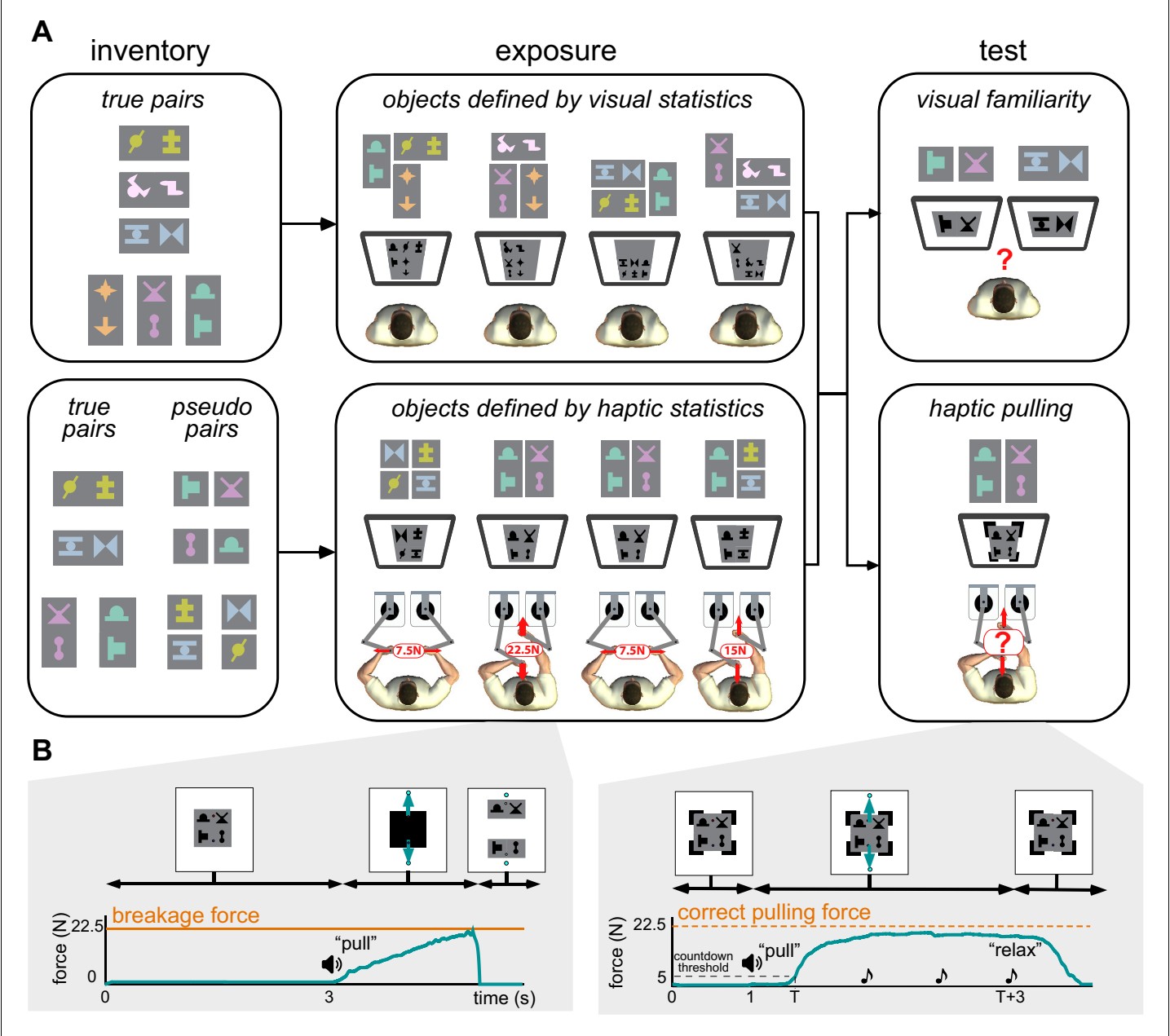

**Figure 1.** Experimental paradigm. (**A**) Main phases of the experiments. Left. An inventory was constructed by arranging abstract shapes into horizontal and vertical pairs. True pairs behaved as objects: their shapes always appeared together, and in the same relative spatial configuration, and were hard to pull apart physically. Pseudo pairs served as controls: they had consistent visual statistics but were as easy to pull apart as two separate objects (indicated by the small separation between their shapes). Colouring and separation for illustration only, participants saw all shapes in grey-scale during exposure and testing, with no gaps between them, so that no visual cues separated the pairs of a compound scene (as shown on screens in the center and right panels). Center. During the exposure phase, participants experienced a sequence of visual scenes showing compound objects consisting of several pairs. The way the image displayed on the screen was constructed from the inventory is shown above each screen in colour for illustration. In the first experiment (top), participants observed compound scenes each constructed from three true pairs of the inventory. In the second experiment (bottom), on each trial, a compound scene consisting of two pairs (true or pseudo) was displayed and participants were required to pull the scenes apart in one of two directions as shown. A bimanual robotic interface (*Howard et al., 2009*) was used so that participants experienced the force at which the object broke apart (breakage force shown in red) but, crucially, visual feedback did not reveal the identity of true and pseudo pairs (see Materials and methods). Thus, only haptic information distinguished the true and pseudo pairs as the force required depended on the underlying structure of the scene. Right. In both experiments, participants finally performed two tests. First, in the haptic pulling test (bottom), participants were asked to pull with the minimal force which they thought would break apart a scene, composed of true or pseudo pairs (in both directions). We measured this force by 'clamping' the scene so that no haptic feedback was provided about the actual breakage force (black clamps at the corners of

*Figure 1 continued on next page*

*Figure 1 continued*

the scene). Crucially, the visual display also did not reveal the identity of true and pseudo pairs. Second, in the visual familiarity test (top), participants were asked to select which of two scenes presented sequentially appeared more familiar. One scene contained a true pair and the other a chimeric pseudo pair. Selecting the true pair counted as a correct response, but no feedback was given to participants as to the correctness of their choices. (B) Timeline of events in haptic exposure and test trials (displayed force traces are from representative single trials). Left. Haptic statistical exposure trials had scenes consisting of combinations of true and pseudo pairs of the inventory (top). After a fixed amount of time, the scene was masked (black square covering the scene), then pulling was initiated ('pull' instruction was played), and the scene was unmasked and shown as separated once the pulling force (green arrows and curve) exceeded the breakage force (orange line). Right. In the haptic pulling test, participants were asked to generate a pulling force which they thought would be just sufficient to break the scene apart (ideally the breakage force corresponding to the scene, orange dashed line). The scenes were constructed using the pairs of the inventory without any visible boundary between them and held together by virtual clamps at the corners of the scene (top). Pulling was initiated ('pull' instruction), and once the participant's pulling force (green arrows and curve) exceeded a 5 N threshold (dashed black line), three beeps were played at 1 s intervals (notes). The clamps remained on until the end of the trial (top), so the scene never actually separated, and after the third beep (at which the pulling force was measured) participants were asked to 'relax'. See Materials and methods for details of the variant used in the haptic exposure task.

DOI: https://doi.org/10.7554/eLife.43942.002

The following figure supplements are available for figure 1:

**Figure supplement 1.** Phases of the visual statistical exposure experiment.

DOI: https://doi.org/10.7554/eLife.43942.003

**Figure supplement 2.** Phases of the haptic statistical exposure experiment.

DOI: https://doi.org/10.7554/eLife.43942.004

were expected to pull harder when the pulling direction was parallel to the orientation of the pairs as this required both objects to be broken in half. Conversely, we expected them to pull less hard

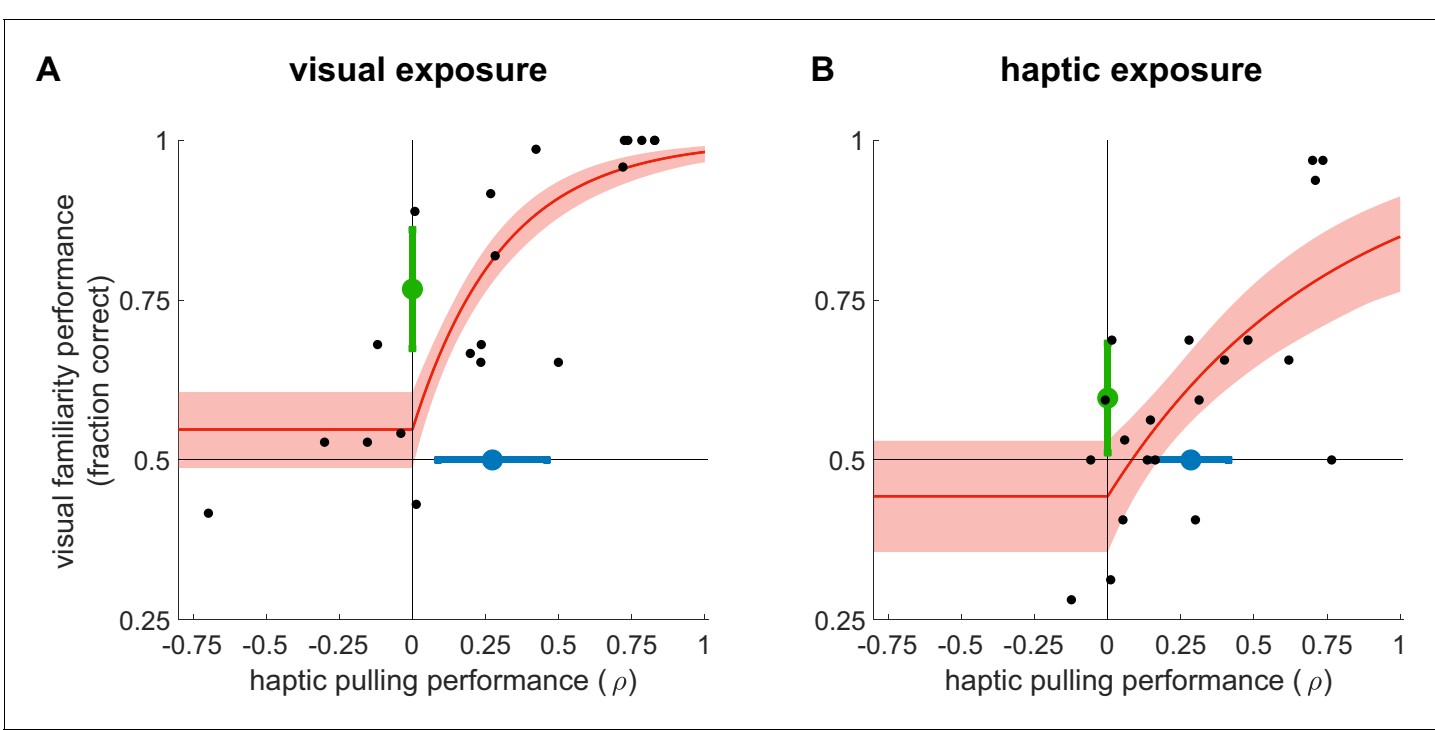

**Figure 2.** Learning from exposure to visual (A) and haptic statistics (B). Performance on the visual familiarity test against haptic pulling performance for individual participants (black dots) with rectified exponential-binomial fit (red ±95% confidence limits, see Materials and methods). Visual familiarity performance was measured by the fraction of correct responses (selecting true over pseudo pairs). Haptic pulling performance was quantified as the correlation coefficient (ρ) between the true breakage force and participants' pulling force across test scenes. Average performance (mean ±95% confidence intervals) across participants in the two tasks is shown by coloured error bars (familiarity: green, pulling: blue). Vertical and horizontal lines show chance performance for visual familiarity and haptic pulling performance, respectively. Note that in the first experiment (A), the performance of two participants was identically high in both tasks, and thus their data points overlap in the top right corner of the plot.

DOI: https://doi.org/10.7554/eLife.43942.005

on trials in which the pulling direction was orthogonal to the orientation of the pairs as this only required them to separate the two objects from each other. We measured participants' performance as the correlation (ρ) between their pulling force and the required breakage force (see *Figure 3* and Materials and methods). Participants performed significantly above chance (*Figure 2A*, black dots: haptic pulling performance for individuals, blue dot and error bar: group average ρ = 0.27 [95% CI:

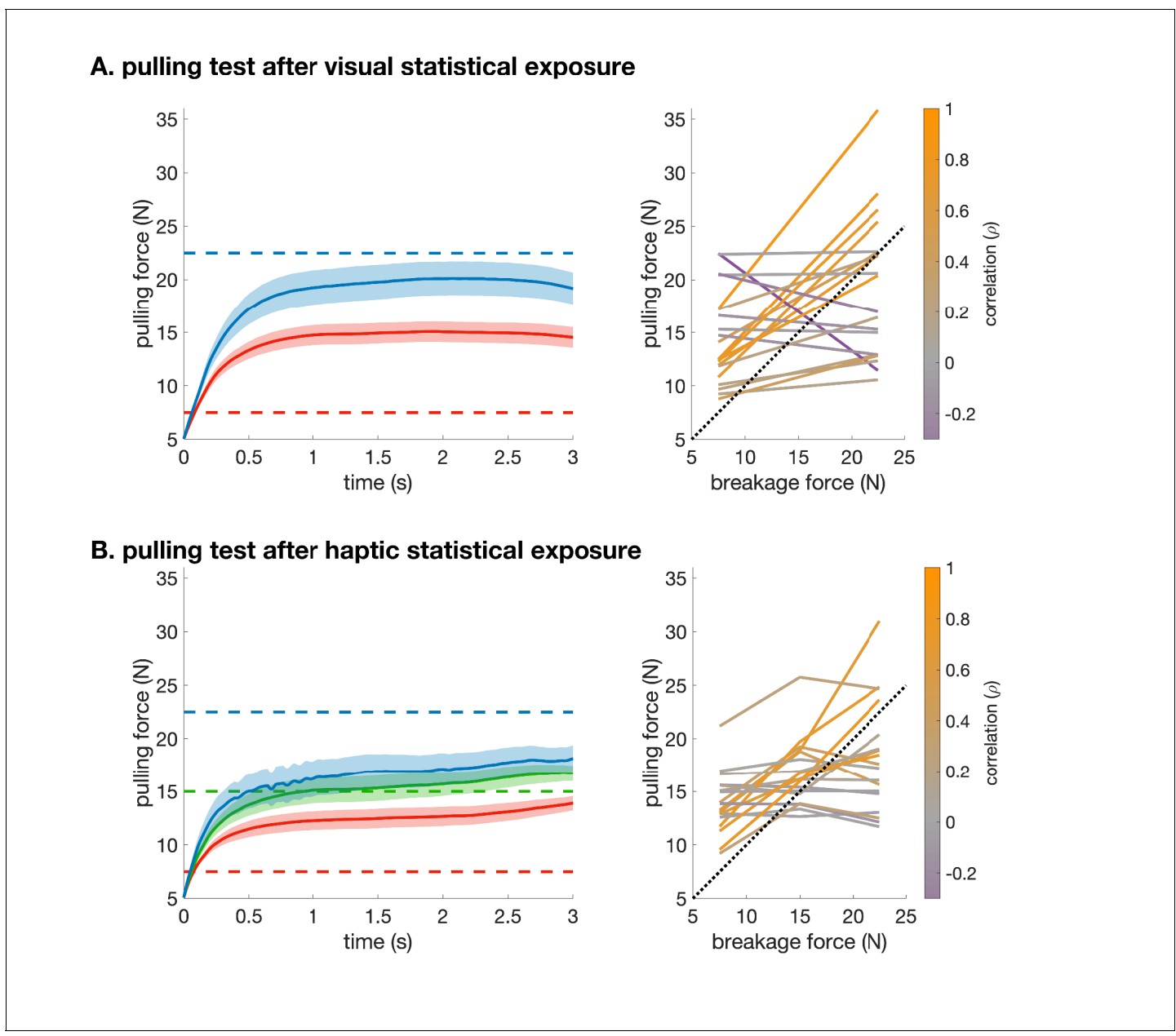

**Figure 3.** Pulling performance in the visual (A) and haptic (B) statistical exposure experiment. (**A**) Left: Force traces from the start of pulling (5 N) on clamp-catch trials in the haptic pulling test of the visual statistical exposure experiment. Data shows mean ± s.e.m. across participants for trials in which the breakage force was high (blue) or low (red). Dashed lines show the corresponding breakage forces. Right: Average pulling force (at 3 s) vs. breakage force (two levels) for each participant colour coded by their correlation (across all trials). Dotted line shows identity. The average pulling force difference between the two levels was 9.3 N ± 3.3 N (s.e.m.). (**B**) as A for the clamp trials in the haptic pulling test of the haptic statistical exposure experiment, in which there were three levels of breakage force. The average pulling force difference between the low and medium breakage force levels was 5.4 N ± 1.2 N (s.e.m.), and between the medium and high breakage force levels was 2.8 N ± 1.7 N (s.e.m.). Raw data necessary to generate this figure was only saved for 18 participants.

DOI: https://doi.org/10.7554/eLife.43942.006

0.07–0.47], $t(19) = 2.88$, p=0.0095, Bayes factor = 5; see also *Figure 3A*). While this effect was weak on average, more importantly, across participants, there was also a highly significant positive relationship between their performance on the visual familiarity and haptic pulling test (*Figure 2A*, red, rectified exponential-binomial fit, see Materials and methods, likelihood ratio test, $\chi^2(2)=265$, $p<1\cdot10^{-10}$, $\log_{10}$ Bayes factor = 141). In particular, our fit of the data revealed that going from random haptic pulling performance ($\rho = 0$) to perfect performance ($\rho = 1$) covered 87% of the possible range of visual familiarity performance (fraction correct from 0.5 to 1.0). We also tested whether there was a significant change in performance throughout the haptic pulling test trials and found no significant trend (p=0.07, Bayes factor = 1, see Materials and methods) suggesting that generalisation immediately appeared and it did not just gradually develop during the test. These results show that participants successfully generalised what they learned through visual statistics to predict the haptic properties of objects, and suggest that variability in performance on both tasks across participants is due to the same underlying cause: differences in how well participants learned the inventory.

In the second experiment, we examined haptic learning and haptic-to-visual generalisation with a different group of $N = 20$ participants (*Figure 1A*, inventory, bottom). As in the previous experiment, in order to familiarise participants with our setup, we pre-trained them on the basic pulling paradigm with coloured rectangles as objects, without any reference to the shapes of the inventory (*Figure 1—figure supplement 2*). They were then exposed to the haptic statistics of the inventory (*Figure 1A*, exposure, bottom; *Figure 1B*, left). During exposure, each scene consisted of 4 shapes arranged as a 2×2 block on a grey square (*Figure 1A*, exposure, bottom), and participants were required to pull apart these scenes in predefined directions so as to part the scene into two equal pieces (*Figure 1B*, left; that is into two vertical pieces for horizontal pulling and vice versa). Again, unknown to the participants, each scene was constructed by combining two of the objects from the inventory (either a pseudo-pseudo, a pseudo-true, or a true-true combination of pairs arranged either vertically or horizontally). We chose to have only 2 and not three objects in each scene so that participants always knew the scene would break apart simply into two pieces — the physics of multiple objects with complicated (potentially non-convex) geometries would have been much more difficult to simulate and expect participants to understand.

Critically, the force (simulated by the bimanual robotic interface) at which each scene separated depended both on the constituent pairs and the pulling direction, and only true pairs behaved haptically as unitary objects in that their shapes required more force to separate than the shapes of pseudo pairs, or shapes belonging to different pairs. This led to three different force levels required to separate the scenes, with the lowest force when pulling apart any combination of pairs orthogonal to their boundary (*Figure 1A*, Haptic exposure: examples 1 and 3, 7.5 N), the highest force required when separating two true pairs into their constituent pieces (2nd example, 22.5 N), and an intermediate force when separating a true and a pseudo pair into their constituent pieces (4th example: 15 N). As participants pulled on each side of the scene, the resistive force generated by the robots rose until it reached a threshold (7.5, 15 or 22.5 N depending on the scene), at which point the forces dropped to zero and the scene parted visually. The shapes were masked from just before pulling started until the scene was successfully separated. Thus, the duration for which the shapes were seen as unseparated and then separated also conveyed no information about the identity of the true and pseudo pairs. Importantly, although these participants had visual experience with the objects, true and pseudo pairs appeared the same number of times and with the same consistency (i.e. their constituent shapes always appeared together in the same spatial configuration), and so visual information could not be used to distinguish between them. Therefore, objects (true pairs) could only be identified by the physical effort required to pull the scenes apart.

Following exposure, participants were tested on the same two tasks as in the previous experiment. Performance on the haptic pulling test showed that participants successfully learned which scenes required more or less pulling force (*Figure 2B*, black dots: haptic pulling performance for individuals, blue dot and error bar: group average $\rho = 0.28$ [95% CI: 0.14–0.42], $t(19) = 4.34$, $p=3.8\cdot10^{-3}$, Bayes factor = 91; see also *Figure 3B*). Haptic experience also affected participants' judgements in the visual familiarity test, in which they needed to compare two pairs, one a true pair and the other a pseudo pair. Participants judged true pairs significantly more familiar than pseudo pairs (*Figure 2B*, black dots: visual familiarity performance for individuals, green dot and error bar: group average quantified by fraction correct 0.6 [95% CI: 0.51–0.69], $t(19) = 2.2$, p=0.038). Note

that this across-modality effect was even weaker on average than previously (Bayes factor = 2 indicates evidence that is weak or not worth mentioning) because, in contrast to the previous experiment, haptic and visual statistics were now in explicit conflict: true and pseudo pairs (compared in the visual familiarity task) were identical in their visual statistics and only differed in their haptic statistics. As there was no haptic stimulus during visual statistical exposure in the other experiment, no such conflict arose there.

More critically, we also found again that participants' familiarity performance had a highly significant positive relationship with their pulling performance (*Figure 2B*, red, rectified exponential-binomial fit, likelihood ratio test, $\chi^2(2)=47.2$, p=$5.6 \cdot 10^{-11}$, $\log_{10}$ Bayes factor = 35), such that performance on the haptic pulling test accounted for 81% of visual familiarity performance. As before, there was no significant change in performance throughout the familiarity test trials (p=0.58, Bayes factor <1, see Materials and methods) suggesting that the generalisation effect did not gradually emerge during the test trials. These results parallel the results of the visual exposure experiment. Moreover, they demonstrate a particularly strong form of generalisation of information acquired through haptic statistics to judging visual properties of objects — at least in those participants who learned the haptic statistics well. That is, objects that appeared precisely the same number of times as others were 'illusorily' but systematically perceived as visually more familiar just because they had more object-like haptic properties. Interestingly, we found similar levels of haptic performance in the two experiments ($t(18) = 0.09$, p=0.93, Bayes factor (favouring the same performance levels)=3, see Materials and methods) even though in the first experiment there was no haptic statistical exposure at all and participants' haptic performance relied only on generalisation from the visual exposure. Performance on the visual familiarity test was higher after visual exposure than after haptic exposure ($t(18) = 2.65$, p=0.01, Bayes factor (favouring different performance levels)=4) which was expected based on the fundamental difference in cue conflicts between the two experiments.

In order to test whether the positive relationship between performance on the two tasks *across* participants (*Figure 2A and B*, red) was not merely due to generic (e.g. attention-based) sources of modulation, we performed a *within*-participant analysis of object-consistency (see Materials and methods). This analysis measured, for each participant, how much the particular pairs they regarded as the true objects of the inventory during the visual familiarity test (and hence indicated as more familiar) were also the ones that they treated as the true objects during the haptic pulling test (and hence pulled harder when needed to break them). This was quantified by a single scalar measure (correlation) between familiarity and pulling force for individual scenes as a measure of consistency (see Materials and methods). As this was a noisy measure, based on a limited number of trials with each participant, we then pooled the data from both experiments and used a t-test across the participants to ask if this measure was significantly different from zero. We found a significantly positive consistency (correlation $0.297 \pm 0.104$ with $t(34) = 2.86$, p=0.007, Bayes factor = 6). Taken together, this demonstrates that participants developed a modality-generic representation of objects from either visual or haptic statistical contingencies alone, which in turn they could transfer to the other modality.

Finally, we tested whether the generalisation between visual and haptic statistics required an explicit sense of knowledge about the shape pairs (*Figure 4*). Quantitative debriefing data were collected from 23 participants after the experiments from which we computed the proportion of correctly identified true pairs as a measure of explicit knowledge (*Figure 4A*). Across these participants, the performance in the visual familiarity and in the haptic pulling test strongly correlated ($R = 0.84$ [95% CI: 0.65–0.93], p=$6.2 \cdot 10^{-7}$, Bayes factor = 3225, see also *Figure 4B*). Critically, when we controlled for participants' explicit knowledge (proportion of correctly identified pairs) on the relationship between visual and haptic performance, we still found a highly significant partial correlation ($R = 0.69$ [95% CI: 0.38–0.86], p=$3.1 \cdot 10^{-4}$, Bayes factor = 23.4, see also *Figure 4C*) suggesting strong implicit transfer between modalities in addition to that afforded by this kind of explicit knowledge. Furthermore, the ratio of the explained variances ($R^2$) shows that the larger part (67%) of the generalisation effect is due to implicit transfer and cannot be explained by explicit reasoning about the pairs.

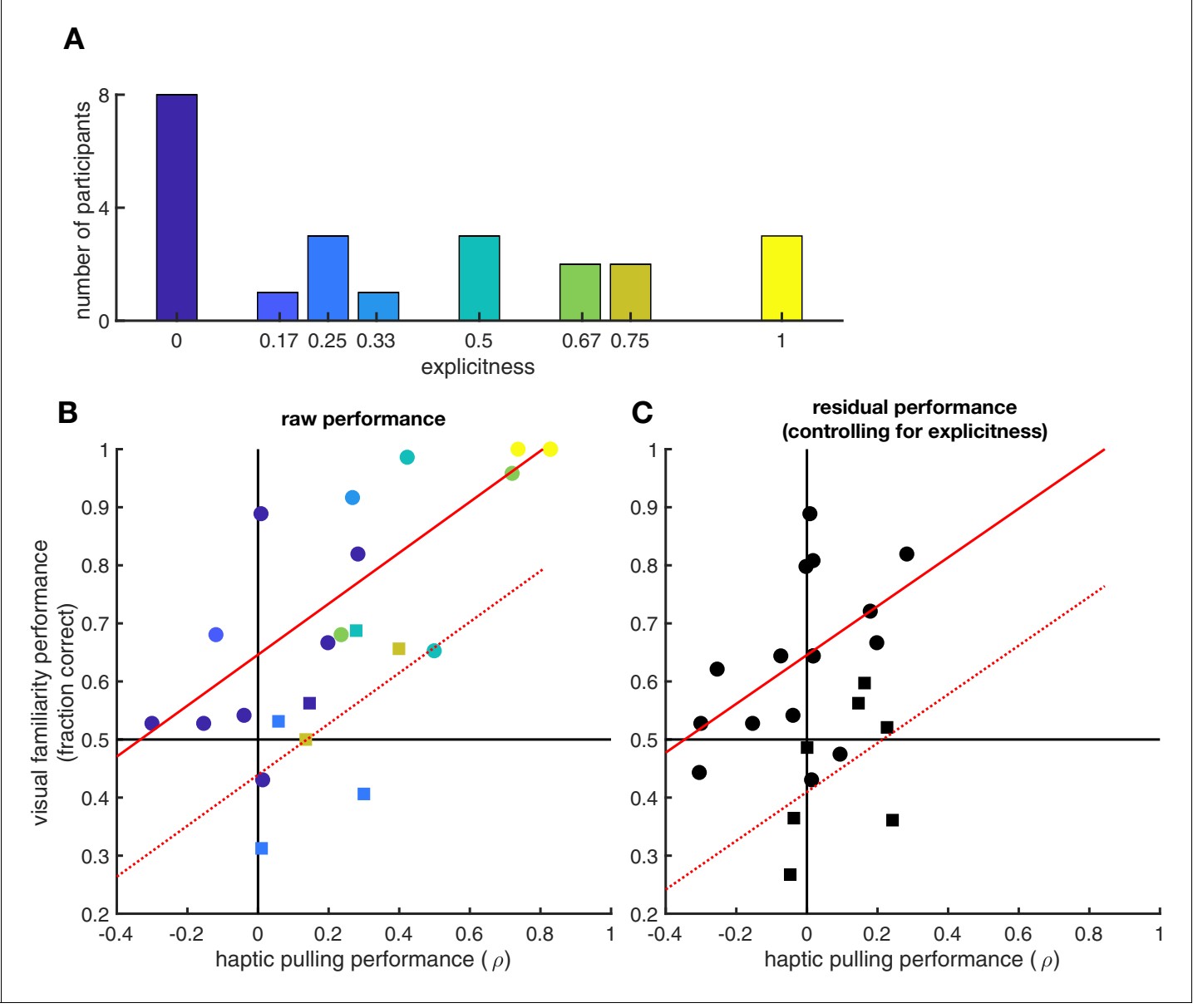

**Figure 4.** Effects of explicit knowledge on generalisation. Participants' explicit sense of knowledge was quantified as the proportion of true pairs they correctly identified (out of 6 in the visual statistical exposure and out of 4 in the haptic statistical exposure experiment, resulting in nine possible unique levels in total, out of which eight were realized) during a debriefing session following the experiment (N = 23 participants). (**A**) Histogram of explicitness across participants (average = 0.37). (**B**) Visual and haptic performance as in *Figure 2*, pooled across the two experiments for those participants who were debriefed (circles: visual statistical exposure, squares: haptic statistical exposure experiment). Colours show explicitness for each participant as in panel A. Red lines show linear regression assuming same slope but allowing for different average performances in the two experiments (solid: visual statistical exposure, dotted: haptic statistical exposure experiment): $R = 0.84$ (95% CI: 0.65–0.93), p=6.2·10$^{-7}$. (**C**) Residual visual and haptic performance after controlling for explicitness (symbols as in panel B). In each experiment, both haptic and visual performance were regressed against explicitness. Residual performances in each modality were then computed by subtracting the performances predicted based on explicitness from the actual performances. Red lines show linear regression as in panel B: $R = 0.69$ (95% CI: 0.38–0.86), p=3.1·10$^{-4}$.

DOI: https://doi.org/10.7554/eLife.43942.007

## Discussion

In summary, we found evidence that participants could segment scenes into objects based on either visual or haptic statistics alone, without any boundaries that could identify the objects. Such learning led to genuinely coherent object-like representations as participants segmented scenes into objects

consistently across the two modalities, independent of the modality in which the statistics of the objects were originally experienced. Our participants' within- and across-modality performance was not perfect as implicit statistical learning over short periods is known to be difficult (*Kim et al., 2009*; *Perruchet and Pacton, 2006*). However, critically, participants who learned well within one modality showed strong generalisation to the other modality (*Figure 2*), beyond what an explicit sense of knowledge of the objects, potentially leading to highly cognitive strategies, would have predicted (*Figure 4*).

Earlier reports in statistical learning only showed that statistical cues can be used for segmentation (in various sensory modalities). However, they typically focused on a single modality (*Conway and Christiansen, 2005*; *Creel et al., 2004*; *Fiser and Aslin, 2001*; *Hunt and Aslin, 2001*) or showed that humans can combine statistical information simultaneously presented in more than one modality (*Conway and Christiansen, 2006*). Critically, these studies did not investigate the 'objectness' of the resulting representations in any way. In particular, they did not test generalisation across modalities and hence could not exclude the possibility that performance in each modality only relied on information presented in that modality alone, without an underlying modality-general object-like representation. Conversely, other studies showed generalisation across visual and haptic modalities, but they used objects which were already fully segmented by low-level boundary cues and as such they could not investigate the role of statistical learning in the emergence of object-like representations (*Yildirim and Jacobs, 2013*). Instead, our findings suggest a deeper underlying integration of object-like representations obtained by statistical learning: any statistically defined structural information obtained in one modality becomes automatically integrated into a general internal representation linking multiple modalities.

Although our experiments were conducted with adult participants, infants have also been shown to learn to segment visual scenes or auditory streams automatically, after mere passive exposure (*Fiser and Aslin, 2002*; *Kirkham et al., 2002*; *Quinn and Bhatt, 2005*; *Saffran and Kirkham, 2018*). Importantly, these studies used stimuli with a statistical structure (and in the case of visual experiments, actual constituent shapes) that were similar or even identical to those used in our experiment (*Fiser and Aslin, 2002*; *Quinn and Bhatt, 2005*; *Saffran et al., 1996*; *Saffran and Kirkham, 2018*). This suggests that, by parsimony, infants possess the same sensitivity to the co-occurrence statistics of sensory inputs as adults (*Aslin, 2017*). Since we showed that statistical learning produces object-like representations in adults, we propose that the statistical learning mechanisms revealed in our experiments might also operate in the emergence of object representations during cognitive development.

If, as we argue, the statistical learning mechanisms we revealed also operate in infants, the present findings complement the results of earlier infant studies on object representations. We tested whether humans can use primarily statistical cues to segment the world into constituent components, which pass a fundamental criterion for objects — that of zero-shot across-modality generalisation (i.e. going beyond observed statistical regularities; *Spelke, 1990*). In contrast, previous studies of cognitive development defined a specific set of criteria, including cohesion, boundedness, rigidity, and no action at a distance, that infants use to identify objects (*Kellman and Spelke, 1983*; *Spelke, 1990*). Our results suggest that these criteria may be *sufficient* but not *necessary*. For example, one might argue that the objects in our experiment violated even the basic requirement of having 3-dimensional structure, and specifically the principle of 'cohesion' of *Spelke (1990)* because their constituent shapes were separated by gaps (although in front of a spatially contiguous grey background). Thus, these classical criteria may be special cases of a more general principle of statistical coherence. Nevertheless, an internal object-*like* representation segmented based on statistical coherence (and other cues) may need to eventually pass a number of additional criteria (e.g. those involving cohesion) to become a real mental object, and it will be for future studies to test whether and how statistical learning mechanisms can produce such representations.

In general, there may be many shades of perceiving 'objectness' (ranging from rigid bodies through more elusive entities, such as a melting scoop of ice cream or the jet of steam of a boiling teapot, to collections of clearly separate objects). Thus, further work will be needed to refine the necessary and sufficient conditions for segmenting entities with different degrees of objectness on this continuum. Similarly, the difference in the behavioural measures used to index object perception in previous studies and in our experiments (looking times vs. across-modality generalisation) may also need more attention. Specifically, it will be interesting to see whether the perception of (a

degree of) objectness is always reflected consistently in all forms of behaviour, or it is subject to paradoxical effects, akin to for example the size-weight illusion (*Flanagan and Beltzner, 2000*), when the same feature seems to be perceived differently in the context of controlling different aspects of behaviour (decision making vs. motor control). From this perspective, our work represents an important first step in connecting the field of statistical learning to the kind of object representations that have been identified in infants.

Finally, although the present study does not provide empirical evidence for a single specific cognitive mechanism underlying the generalisation effects we found, these results together with previous studies (*Lake et al., 2015*; *Orbán et al., 2008*) point to possible computations explaining the present findings. First, the generalisation effects occurred without any ancillary cues that are required to engage specialized learning mechanisms, such as segmentation cues for implicit rule learning (*Peña et al., 2002*; *Saffran et al., 2007*), verbal instructions for explicit hypothesis testing (*Shanks, 2010*), or ostensive signals for social learning (*Csibra and Gergely, 2006*). Second, it is unlikely that participants were able to retain in memory all the raw sensory stimuli they received during the exposure phase (e.g., 444 scenes with six shapes in each for the statistical exposure experiment). Thus, they must have developed some compressed representation of those stimuli during exposure, and it is only this representation that then could allow them to generalise in the test phase. Third, with regard to the form of the compressed representation, statistical learning goes beyond the learning of simple (pairwise) associations between the constituent components of objects, and has been shown to be best described as the extraction of statistically meaningful (potentially multivariate) latent 'chunks' (*Gershman and Niv, 2010*; *Orbán et al., 2008*; *Yildirim and Jacobs, 2012*). Therefore, we propose that these latent chunks are the abstract representations that are built automatically during exposure and mediate the across-modality effects we observed. Accumulating evidence supports this view by showing that the neural representation underlying multimodal integration might involve cortical areas traditionally linked to unimodal processing (*Amedi et al., 2001*; *Ghazanfar and Schroeder, 2006*). Together, these results suggest that statistical learning is not only a domain-general mechanism (*Frost et al., 2015*; *Kirkham et al., 2002*; *Thiessen, 2011*) but it also results in domain-general internal representations that could be the basis for the emergence of affordances (*Gibson, 1979*; *Parker and Gibson, 1977*) and the abstraction of object concepts (*Carey, 2011*; *Leslie et al., 1998*; *Spelke, 1990*).

## Materials and methods

### Participants

In the visual statistical exposure experiment, 28 participants (age range 19–39, mean 25, 20 women) gave informed consent and participated. Eight participants were excluded from full analysis as they did not achieve significant performance in haptic task training (see below, 'Exclusion criteria'). The final sample therefore comprised of 20 participants (age range 20–39, mean 25, 15 women).

In the haptic statistical exposure experiment, 20 participants (age range 21–34, mean 26, 16 women) gave informed consent and participated. No participants were excluded (see below, 'Exclusion criteria').

Data was collected in two instalments. First, we made a preliminary estimate of the approximate number of participants we would need for significant results and collected data accordingly. This resulted in 23 and 13 participants in the two experiments (16 and 13 after exclusion criteria were applied). All our main results (relationship between visual and haptic performance in each experiment) were highly significant and resulted in Bayes factors > 10. Subsequently, an external expert not involved either in the design of the study or in the analysis of the data, or invested in the success of our study, suggested that data from 20 participants in each experiment should be collected. Therefore, we collected data from additional participants to reach 20 participants after exclusion in each experiment. Again, all our main results remained highly significant. Thus, the process of adding participants, and the consistent usage of Bayes factors throughout (see below), ensured our study was not biassed towards favourable results (*Dienes, 2011*).

All experimental protocols were approved by the University of Cambridge Psychology Ethics Committee.

## Equipment

We used two vBOTs to provide haptic stimuli and record haptic responses (during haptic exposure and testing, respectively). The vBOT is a custom-built back-drivable planar robotic manipulandum exhibiting low mass at its handle, with the handle position measured using optical encoders sampled at 1000 Hz, and torque motors allowing endpoint forces to be specified (*Howard et al., 2009*). Participants were seated in front of the apparatus and held one vBOT handle in each hand. By using two horizontally adjacent vBOTs, we applied haptic stimuli and recorded responses bimanually (*Figure 1A*, Haptic exposure, and Haptic testing). Visual stimuli were displayed using a computer monitor projected to the participant via a horizontal mirror mounted above the vBOTs (*Figure 1A*, Haptic exposure, Visual exposure, Haptic testing, and Visual testing). During haptic exposure and testing, the participants' veridical hand positions were represented using two cursors (0.3 cm radius) overlaid in the plane of the movement. Responses during visual testing were recorded by closure of the switches on the vBOT handles.

## Visual stimuli

In both experiments, following previous work (*Fiser and Aslin, 2001*), visual stimuli for statistical learning consisted of 12 (visual statistical exposure experiment) or 8 (haptic statistical exposure experiment) black abstract geometric shapes of intermediate complexity arranged along a regular grid (without grid lines shown) on a grey background (*Figure 1A*). Unbeknownst to participants, the shapes were grouped into 'true pairs' (the 'objects'), such that constituent shapes of a pair were always shown together and in the same spatial (horizontal or vertical) arrangement, and each shape was part of only one true pair (*Figure 1A*, Inventory, True pairs, coloured only for illustrating pair identity).

The inventory of shapes that could be used for constructing visual scenes included three horizontal and three vertical (visual statistical exposure experiment) or two horizontal and two vertical such true pairs (haptic statistical exposure experiment) as well as an equivalent number of 'pseudo pairs'. The pseudo pairs re-used the shapes of the true pairs such that each horizontal (vertical) pseudo pair consisted of two shapes belonging to two different vertical (horizontal) true pairs, one of them being the top (left) the other the bottom (right) shape of its original true pair (to avoid accidental constellations that could have appeared using true pairs), and each shape was part of only one pseudo pair (*Figure 1A*, Inventory). The assignment of shapes to true and pseudo pairs was randomized across participants to control for effects due to specific shape configurations.

During visual exposure, only true pairs and no pseudo pairs were shown. During haptic exposure, pseudo pairs were displayed with the same visual statistics as true pairs (but they differed in their haptic properties, see below). Each visual scene during exposure (and haptic testing) contained several pairs (three for visual exposure, and two for haptic exposure and testing) in juxtaposition, in a non-occluding manner, without any border lines separating them. Each visual scene during the visual familiarity test consisted of a single true or pseudo pair.

In general, note that the co-occurrence statistics relevant for statistical learning of pairs included both the number of times two shapes appeared together and the number of times each appeared alone (see a formal definition in *Orbán et al., 2008*). This meant that true pairs had stronger overall statistical contingencies in the visual than in the haptic statistical exposure experiment as shapes of a true pair never appeared without each other in the former while they did in the latter due to pseudo pairs.

## Controlling for special cues

Crucially, the instructions to the participants did not refer to the existence of objects in any way, only that there were visual or haptic patterns they needed to observe (see also below). We controlled the stimuli for low-level visual segmentation cues, such that there were no boundaries or colour differences revealing the objects present in a scene (the colour coded shapes in *Figure 1A* illustrate the construction of scenes but these were never displayed to participants). Moreover, while individual shapes were clearly separated, the separation between adjacent shapes belonging to the same or different objects was the same and thus uninformative as to object boundaries. Therefore, objects were not defined, as might naively be expected from observing individual scenes, at the level of a single shape or all shapes in a scene. Instead, the only information available to identify the

objects was the statistics of either their visual co-occurrences or of the physical interactions they afforded across the exposure scenes.

## Visual statistical exposure experiment

The experiment consisted of four phases: (1) visual statistical exposure, (2) haptic task training, (3) haptic pulling test, and (4) visual familiarity test.

(1) Visual statistical exposure. Participants were exposed to a series of scenes constructed using an inventory of 12 shapes arranged into six true pairs, each scene being composed of 3 pairs arranged along a 3-by-3 grid (*Figure 1A*, Visual exposure, top coloured pairs shows the construction and below the screen display, see also above). To ensure there were no obvious spatial cues identifying individual pairs, the positions of the pairs within the grid were randomized with the following constraints: (1) at least one shape in a pair had to occupy a grid location adjacent to one shape in another pair, (2) the central square needed to be occupied by a shape, and (3) the exact same configuration of 3 pairs but at a different location on the grid were considered the same. These spatial constraints generated a set of 444 unique scenes, in which each of the 6 pairs appeared 222 times (see *Figure 1—figure supplement 1*). The scenes were generated as follows (where H and V are horizontal and vertical pairs, respectively):

- 3H gives
  3 (identity of H1) × 2 (identity of H2) × 1 (identity of H3) × [1 (all aligned) + 2 (left/right displacement of one H) × 3 (which H is displaced)] = 42
- 2H and 1V with the two H aligned gives
  3 (identity of H1) × 2 (identity of H2) × 3 (identity of V) × 2 (location of V on left/right) × [1 (V aligned) + 2 (V offset top/bottom)] = 108
- 2H and 1V with the two H offset gives
  3 (identity of H1) × 2 (identity of H2) × 2 (location of H1 offset left/right) × 3 (identity of V) × 2 (location of V above/below) = 72
  Total = 222
  And the same for 3 V and 2V and 1H, giving a total of 444 scenes.

These scenes were presented in a pseudorandom order one at a time for 700 ms, with 1 s pauses between them, and participants were instructed to simply view the scenes without any explicit task other than paying attention so that later they could answer simple questions related to the scenes (Visual familiarity test, see below). The instructions simply asked participants to 'observe each display carefully so that you can answer simple questions related to the pattern of symbols that you observed in all of the displays'.

(2) Haptic task training. Before haptic testing, participants completed a haptic training task on the vBOT in order to familiarise them with the forces associated with pulling apart objects in different configurations (*Figure 1—figure supplement 1*). Each scene consisted of two 2-by-1 rectangles (the 'objects', both being either horizontally or vertically oriented) touching on their long sides, so that the configuration was a 2-by-2 block of coloured pieces. In order to avoid any ambiguity about object boundaries in this case, the identity of the two rectangles was clearly revealed by the different colours of the two rectangles (four colours were used in total). After each scene appeared on the screen, the two vBOTs moved the participant's hands passively to circular placeholders on opposite sides of the scene (either vertically or horizontally, chosen randomly). After a period of time (3 s for *Standard trial*s and 1 s for other trial types, see below) the participant was instructed to pull the object apart (computer generated speech 'pull') in this predetermined direction. Haptic feedback was provided by simulating a stiff spring (spring constant 30 N/cm) between the handles with a length set to 16 cm corresponding to the initial hand separation (see below). On the next trial, the pulling procedure was repeated with the orthogonal pulling direction with the same scene, after which the next scene was generated. The training consisted of three trial types: Standard, Clamp and Clamp-catch trials.

On Standard trials, as participants increased their pulling force against the simulated stiff spring, the object broke apart both haptically (force reduced to zero) and visually (split in the direction corresponding to the pull direction) at a predetermined force threshold. Crucially, the threshold at which the scene broke apart depended on its configuration, and in particular whether the pre-set pulling direction required the breaking of objects (pulling direction parallel to the orientation of the

objects, and hence to the boundary between them) or not. Specifically, the threshold pulling force was determined for each scene by simulating forces between individual pieces such that pieces belonging to the same object were attached by 11.25 N, and pieces belonging to different objects were attached by 3.75 N. This meant that pulling two objects apart without breaking them required a low force (7.5 N) whereas breaking each object into two required a high force (22.5 N) (*Figure 1—figure supplement 1*).

Clamp trials were identical to Standard Trials except that the objects were held together initially by virtual clamps displayed at the four corners. Once the participant started to pull (pulling force exceeded 5 N), three tones were played at 1 s interval and participants were asked to generate the minimal force which they thought would break the scene apart by the final tone. The clamps then disappeared and the scene separated if the force threshold was exceeded. Otherwise, participants were instructed to increase their pulling force until the scene separated.

Clamp-catch trials were similar to Clamp trials except that after the final tone the clamps remained and participants were instructed to 'relax' (stop pulling) so that the scene did not actually break apart on these trials and no feedback on the accuracy of the participant's pulling force was given.

Participants were exposed to a total of 56 trials: 24 Standard trials, followed by 16 Clamp trials, and finally 16 Clamp-catch trials.

(3) Haptic pulling test. This test followed a similar format to the haptic task training but using the true pairs of the original shape-inventory as objects, which thus had no visible boundary between them, and only Clamp-catch trials (i.e. no feedback on the accuracy of their pulling force was ever given, and no scenes were ever separated, see above). Visual scenes with a 2×2 block of four shapes were displayed such that two pairs with the same orientation (both horizontal or both vertical) were chosen randomly from the set of all true pairs. Participants were presented with 48 scenes in total (2 × 24-trial blocks). Within each 24-trial block each combination of two true pairs of the same orientation was presented twice, once for horizontal and once for vertical pulling (the order of scenes was randomly permuted within each block). Note that while this phase did not provide haptic experience with the objects, it did provide additional visual statistical information, in somewhat simpler scenes (2 rather than three objects in each) but still without boundaries, so for these purposes it could be regarded simply as extra visual familiarisation.

(4) Visual familiarity test. Lastly, participants performed a sequence of two-alternative forced choice trials. In each trial, they had to indicate which of two consecutively displayed scenes was more familiar. Scenes were presented sequentially for 1 s with a 1 s pause between them. One of the scenes contained a true pair, the other a pseudo pair of the same orientation. Horizontal pseudo pairs were generated from the shapes of vertical true pairs while the vertical pseudo pairs were generated from the shapes of horizontal true pairs. Participants selected which pair was more familiar by closing the switch on the left (1 st pair) or right (2nd pair) vBOT handle. Participants performed 72 trials (2 × 36-trial blocks) in total. Within every 36-trial block each true pair was compared to each pseudo pair of the same orientation in each order exactly once (the order of trials was randomly permuted within each block). Note that only in this last phase did participants see individual, separated objects (constructed from the shapes of the inventory), of which the boundary was thus obvious.

## Haptic statistical exposure experiment

The experiment consisted of four phases: (1) haptic task training, (2) haptic statistical exposure, (3) haptic pulling test, and (4) visual familiarity test. Note that the ordering of the main phases of the experiment (statistical exposure → haptic testing → visual testing) remained identical across the two experiments (*Figure 1A*). However, the ordering of the haptic task training phase was chosen so that it immediately preceded that phase of the experiment in which haptic experience was first combined with the shapes of the inventory, that is the haptic statistical exposure phase in this experiment and the haptic pulling test in the visual statistical exposure experiment.

(1) Haptic task training. This was similar to the haptic task training in the visual statistical exposure experiment, except that scenes could include not only two differently coloured 2-by-1 rectangles as before (C2: that is two colours) but also one 2-by-1 rectangle and two 1-by-1 squares (C3), or four 1-by-1 squares (C4, *Figure 1—figure supplement 1*). All these configurations were arranged in a 2-by-2 block of coloured pieces as before, and as all 'objects' (rectangles or squares) were differently coloured they still had clear, visually identifiable boundaries between them (four colours used in

total). The additional configurations were needed as the haptic statistical exposure included pseudo as well as true pairs, where pseudo pairs behaved haptically as two separate single elements, and so three rather than two force levels were possible [see below] which thus needed to be demonstrated during haptic task training. The required minimal pulling forces were determined as above. This meant that the same two force levels (7.5 and 22.5 N) were needed to pull apart C2 scenes in the easy (orthogonal to the boundary between the rectangles) and hard directions as in the visual statistical exposure experiment (see above), while C4 scenes were easy (7.5 N) to pull apart in either direction, and C3 scenes were easy (7.5 N) to pull apart in the direction orthogonal to the long side of the rectangle and medium hard (15 N) in the other direction.

Participants completed a total of 144 trials, which consisted of 96 Standard trials (composed of two 48-trial blocks), followed by 48 Clamp trials. (Clamp-catch trials were omitted as it was not necessary to include them in the haptic pulling test, see below.) Trials within each block consisted of 8 trials with C2, eight trials with C4 and 32 trials with C3 in a pseudorandom order and orientation. These proportions were chosen to match those used in the haptic statistical exposure phase, see below.

(2) Haptic statistical exposure. This phase was similar to the haptic test in the visual statistical exposure experiment but included both true and pseudo pairs, and only Standard trials. Specifically, each visual scene could be composed of either two true pairs, or a true pair and a pseudo pair, or two pseudo pairs, such that the two pairs always had the same orientation, touching on the long side, thus forming a 2×2 block of four shapes without a visible boundary between the pairs. Critically, pseudo pairs were indistinguishable from true pairs based on their visual appearance statistics: they appeared the same number of times, in the same combinations with the other pairs. This was important so that any consistent preference in the visual familiarity test (see below) between true and pseudo pairs could have only been due to their different haptic statistical properties. Specifically, pseudo pairs behaved haptically as two separate single-shape objects, rather than one integrated object, so that the constituent shapes of a pseudo pair were as easy to pull apart as shapes belonging to two different objects. This meant that three force levels were required: two true pairs were hard (22.5 N) to pull apart in the direction parallel to their boundary and easy (7.5 N) in the other direction, two pseudo pairs were easy (7.5 N) to pull apart in either direction, and a true and a pseudo pair was easy (7.5 N) to pull apart in the direction orthogonal to the long side of the two pair and medium hard (15 N) in the other direction.

In order to ensure that the time for which each scene was presented in an unseparated and separated state was independent of how much time participants spent on pulling it apart, in each trial, the 2×2 block of four shapes was masked 3 s after the hands were moved into their home positions (i.e. just before pulling could start) and unmasked once the scene was successfully separated. Note that according to the rules of the task (see above) all trials ended by the scene eventually becoming separated, regardless of its composition and the pulling direction. Thus, the visual statistics of the scenes remained independent from their haptic properties and conveyed no information about the identity of the true and pseudo pairs. The instructions simply told participants that 'the force required to break the block apart in each direction will depend only on the symbols and their configuration on the block' and asked them to 'learn the minimal force required to pull the block apart in each direction and we will test you on this later'. (Note that, in contrast to other phases of the experiment involving haptic manipulations, no Clamp or Clamp-catch trials were needed in this phase as we were only exposing participants to haptic statistics but not yet measuring their performance — which occured in the next phase, the haptic pulling test, see below.)

Participants completed 192 Standard trials (composed of four 48 trial blocks). Each block of trials included all possible combinations of two pairs of the same orientation in both pulling directions. This meant that scenes with two true pairs were presented eight times (4 × 22.5 N trials and 4 × 7.5 N trials), scenes with two pseudo pairs were presented eight times (8 × 7.5 N trials) and scenes with a true and a pseudo pair were presented 32 times (16 × 7.5 N trials, 16 × 15 N trials). Trials within each block were randomized.

Note that there were fewer trial scenes in the haptic than in the visual statistical exposure experiment because less unique scenes could be generated in the 2×2 arrangement. Moreover, due to the time the robotic interface needed to shift from one pulling position to the other, the presentation time of the scenes was longer in the haptic exposure than in the visual statistical exposure experiment (*Figure 1—figure supplement 2*).

(3) Haptic pulling test. In order to measure how much participants learned from haptic statistical exposure, we tested their haptic performance as in the other experiment. Therefore, this phase was similar to the haptic pulling test in the visual statistical exposure experiment, but included both true and pseudo pairs as did the haptic statistical exposure phase of this experiment, and used Clamp trials rather than Clamp-catch trials. (Clamp-catch was unnecessary here as there was no need to prevent participants gaining additional haptic information from these trials in this experiment.) Again, to ensure that each scene could be seen in an unseparated and separated state for a fixed amount of time, irrespective of its haptic properties, it was masked during the period between the removal of the clamps and the separation of the scene.

Participants completed one block of 48 Clamp trials which were similar to one block of Standard trials in the haptic statistical exposure phase, except for the presence of the clamps.

(4) Visual familiarity test. These trials were identical to the familiarity test in the visual statistical exposure experiment. Participants completed 32 trials (2 × 16-trial blocks), such that within every 16-trial block each true pair was compared to each pseudo pair of the same orientation in each order exactly once (the order of trials was randomly permuted within each block). Again, this last phase of the experiment was the first time participants saw individual, separated objects (constructed from the shapes of the inventory), of which the boundary was thus obvious.

Although the assignment of shapes to objects (pairs) was randomized across participants, we found at the end of the experiments that some participants had the same order of trials due to a coding error. Specifically, in the haptic statistical exposure experiment, two participants shared the same haptic exposure sequence. In the visual statistical exposure experiment, three participants shared the same haptic testing sequence and two shared the same visual familiarity testing sequence. There is no reason to believe that the order of trials would affect learning or performance.

## Debriefing

Occasional perfect (100%) performance on the visual familiarity task and informal debriefing with the first batch of participants suggested that some might have been developing explicit knowledge of the pairs. Therefore, we chose to perform quantitative debriefing for the final 23 participants at the end of the experiment (16 in the visual and seven in the haptic statistical exposure experiment). Participants were asked 'Did you notice anything about the shapes during the exposure phase of the experiment?'. If they said 'yes' then they were asked 'What was it that you noticed about the shapes?' and if they said something about pairs, they were shown the inventory of shapes separated on a page and instructed: 'Point to all the shapes that form part of pairs that you remember.' Participants were free to indicate as many pairs as they wanted, and if they identified less than the number of true pairs in the inventory, they were not required to guess the remaining pairs. The eight participants who did not notice any pairs were given an explicitness score of 0, while the other 15 participants correctly identifying at least one pair were given an explicitness score equal to the number of correctly identified pairs divided by the total number of true pairs in the inventory. Thus, the score was only based on correctly identified pairs and we ignored incorrect pairs so as to err on the side of increased explicitness in our measure.

Note that this measure of explicit learning not only required that participants had explicit knowledge of the pairs but also that they had an explicit 'meta-cognitive' sense for this knowledge. It could have been possible that some would have identified some pairs even without having an explicit sense that they did, but note that our visual familiarity task already tested their knowledge of pairs by a two-alternative forced choice familiarity judgment (typically taken as an index of implicit learning) and this additional debriefing at the end of the experiment instead served to rule out that highly cognitive operations accounted for all across-modality generalisation.

## Data analysis

### Basic performance measures

Familiarity trials provided binary data, in which choosing the true pair counted as a correct response. As a summary measure of familiarity, we calculated the fraction correct across all trials for each participant. In haptic trials, we recorded the position and force generated by the vBOTs at 1 KHz. Responses in pulling test trials provided the pulling force generated by participants on the final

beep after 3 s (*Figure 1—figure supplements 1* and *2*). The vBOTs are limited to generating a maximum pulling force of 40 N and therefore pulling forces were clipped at 40 N and this happened on 0.21% of both haptic clamp trials in the visual and haptic statistical exposure experiments, respectively. As a summary measure of the pulling test performance, we calculated the correlation (ρ) between the pulling force and the breakage force across all trials. This measured how much their pulling force aligned with the required breakage force while being insensitive to an overall mismatch in the scale or offset of forces. The only critical feature for our hypothesis was that participants should pull harder to separate true pairs into two, compared to pseudo pairs or junctions between pairs, and the correlation measure with breakage force reflects this feature. (Similar results, not shown, were obtained by using the slope of the correlation instead, which takes into account the scale of forces, but remains insensitive to the reliability with which participants generate their forces.) Even though in the first experiment (objects defined by visual statistics), only two levels of breakage force were possible, we still used correlation to keep our results comparable with the second experiment (with three levels of breakage force). Nevertheless, note that in this case, the correlation, ρ, was also monotonically related to the sufficient statistic, $t$, that a direct a comparison (t-test) of the pulling forces at the two breakage force levels would have used (with the same number of trials at each): $t^2 \propto \rho^2 / (1-\rho^2)$.

Participants' performance on the haptic pulling and the visual familiarity tests were compared across the two experiments with t-tests. In both generalisation tests (haptic pulling test in the visual statistical exposure experiment, and visual familiarity test in the haptic statistical exposure experiment) participants completed two blocks of the same test trials (in a different randomization, see above). In order to test whether there was a significant change in performance throughout the test trials, we compared the performance in the first and the second block using a paired t-test.

## The rectified exponential-binomial model

For each experiment, we fit a rectified exponential-binomial model to predict participants' visual familiarity performance (fraction correct, $f_c$) from their haptic pulling performance (correlation, ρ). This model was not intended to be a mechanistic model of how participants solved the tasks but as a phenomenological model capturing the main aspects of the data. Specifically, it captured three intuitions given our hypothesis that behaviour on the two tasks was driven by the same underlying representation of objects. First, performance on both tasks should depend on how well a participant acquired the inventory of objects through experience in the exposure phase of the experiment, and this common cause should cause co-variability with a monotonically increasing (positive) relationship between the two performance measures. As $f_c$ is upper bounded at 1, we chose a saturating exponential function to parametrise this relationship. Second, participants with chance or below-chance haptic performance (ρ ≤ 0) should have learned nothing about the objects and therefore would have a baseline visual familiarity performance which is independent of ρ. This baseline could in principle be above chance, especially in the visual statistical exposure experiment where participants learn the visual statistics but do not generalise to the haptic domain. Third, performance on individual trials was statistically independent, given the strength of the object representation of the participant. The rectified exponential-binomial is a three-parameter model that captures these intuitions:

$$f_c = \frac{T_c}{T}, \text{where } T_c \sim \text{Binomial}(P(\rho), T), \text{and } P(\rho) = \begin{cases} \beta_0 & \text{if } \rho \leq 0 \\ \beta_1 + (\beta_0 - \beta_1)\, e^{-\rho/\lambda} & \text{otherwise} \end{cases} \quad (1)$$

where $T_c$ is the number of correct and $T$ is the total number of trials in the visual familiarity test ($T$ = 72 and $T$ = 32 for the two experiments, see above), $\beta_0$ and $\beta_1$ determine the range of $f_c$, and $\lambda$ controls the rate of rise of the exponential. We used a likelihood ratio test to examine the null hypothesis that there was no relation between fraction correct and correlation {H0: $\beta_1 = \beta_0$ and thus $\lambda$ has no effect}. In order to compute confidence intervals around the maximum likelihood fits (solid red lines in *Figure 2*), we used the 'profile likelihood' method (*Venzon and Moolgavkar, 1988*). That is, the $1-\alpha$ confidence region encloses all parameters values for which the log likelihood is within $\chi^2_{1-\alpha}(n)/2$ of the maximum log likelihood, where $n$ is the number of parameters being estimated via the method of maximum likelihood (Appendix A in *McCullagh and Nelder, 1989*). Briefly, we sampled 100,000 parameter sets from the Laplace approximation of the log-likelihood (i.e. a Gaussian approximation centred on the maximum likelihood parameter set, with the inverse

covariance determined by the local Hessian of the log-likelihood; *Bishop, 2007*). We rejected those samples for which the negative log-likelihood fell from the maximum by more than $q/2$ where $q$ was the 95th quantile of the $\chi^2$ distribution with 3 degrees of freedom. We then estimated the 95% confidence of the maximum likelihood fit as the extrema of the predictions obtained from the remaining parameter set samples (shaded red regions in *Figure 2*).

We also computed the Bayes factor to directly compare the two hypotheses: 1. that there was a systematic relationship between visual and haptic performance as predicted by the rectified exponential-binomial (*Equation 1*), and 2. the null hypothesis, that is that there was no relation between visual and haptic performance (see also above). This was computed as the ratio of the (marginal) likelihoods of the two hypotheses each of which was approximated as the likelihood evaluated at the maximal likelihood parameter set divided by the square root of the log-determinant of its local Hessian (ignoring constant factors that cancelled or did not scale with the number of data points; *Bishop, 2007*). This is a more accurate approximation of the marginal likelihood than the often used Bayesian information criterion, as it uses information about the Hessian which was available in our case, see also above.

## Within-participant object-consistency analysis

In order to test whether the correlation between performance on the two tasks *across* participants we found (*Figure 2*, red) was not merely due to generic (e.g. attention-based) co-modulation effects, we performed a *within*-participant analysis of object-consistency. In particular, if correlation between performance in the two tasks is really driven by a unified underlying object representation, then the same pairs that participants regard as the true objects of the inventory during the visual familiarity test (and hence indicate as more familiar) should also be the ones that they treat as the true objects during the haptic pulling test (and hence pull harder in the direction parallel to their boundaries). Note that this reasoning is independent of the actual inventory that was set up in the experiment (*Figure 1A*, inventory), and focuses on participants' internal representation, regardless whether it matched the actual inventory or not, only requiring that they behave consistently according to that internal representation in both tasks. In other words, this analysis is able to differentiate systematic deviations in participants' behaviour due to a misrepresentation of objects from errors due to not having proper object-like representations.

To measure object-consistency within a participant, we calculated a haptic and a familiarity score for each unique scene that contained two true pairs in the haptic pulling test (12 and 4 in the visual and haptic exposure experiments, respectively), and computed the correlation between these scores across scenes. The haptic score was the average difference (across the repetitions of the same scene) in the pulling force participants generated when pulling to separate each of the two pairs into two compared to the pulling force generated to separate the two pairs from each other. The familiarity score was the average of the fraction of trials that the participant chose each of the pairs making up the scene as more familiar than another pair in the familiarity test. This score ranges from 0 (they never selected either pair in the familiarity test) to 1 (they always selected both pairs). We performed a t-test on these correlations across all participants, combining across experiments for statistical power. Participants who had a familiarity fraction correct of 1 (5 participants in the visual statistical exposure experiment) were excluded from this analysis as their object consistency-correlation was undefined.

## Controlling for explicit knowledge of pairs

We also tested whether the generalisation between visual and haptic statistics required explicit sense of knowledge about the shape pairs. First, in order to quantify participants' explicit knowledge about the inventory, we computed the proportion of correctly identified true pairs (and ignored incorrectly identified pairs) based on the debriefing data (see Materials and methods, Debriefing). As there were six true pairs in the visual, and four in the haptic statistical exposure experiment, the resulting scores were multiples of 1/6 or 1/4 for the two experiments, respectively (these were combined in *Figure 4*). Next, we computed the correlation between their performance in the visual familiarity and in the haptic pulling test across the two experiments using multiple regression on visual performance with the two covariates being haptic performance and an indicator variable for the type of the experiment (thus allowing for the average performances to depend on the

experiment, but assuming that the regression slope was the same). Finally, partial correlations were measured between the performances in the two tests controlling for the proportion of correctly identified pairs (our measure of participants' explicit knowledge, see above). Partial correlation can reveal whether there is a significant relationship between the visual and haptic performance that cannot be explained by the explicit knowledge of the shape pairs. Specifically, in each experiment, both haptic and visual performance were regressed against explicitness. Residual performances in each modality were then computed by subtracting the performances predicted based on explicitness from the actual performances. The correlation between these residual performances across the two experiments was computed as for the raw performances and yielded our partial correlation measure. We also computed the ratio of the explained variances ($R^2$) of the normal and partial correlations in order to measure the extent to which the generalisation effect could be explained by implicit transfer rather than by explicit knowledge.

## Bayesian tests

In all statistical analyses we computed both the classical frequentist and the corresponding Bayesian tests. We used scaled JZS Bayes factors in the Bayesian t-tests, and in the Bayesian multiple linear regression for the correlational analyses with a scaling factor equal to $\sqrt{2}/2$ in the prior distribution (*Morey and Rouder, 2011*).

## Exclusion criteria

In order to interpret the haptic pulling performance and its relation to visual familiarity performance (see above), it was essential that participants understood the general rules of pulling and scene breakage in our set-up (i.e. that objects were harder to break than to separate), which were used in all haptic task phases (haptic task training, haptic statistical exposure, and haptic pulling test). In the visual statistical exposure experiment, the only indicator of whether participants understood the rules of pulling was their performance on haptic task training. Thus, in this experiment, participants were only included for further analysis if they had a significant ($p<0.05$) positive correlation between their pulling force and the required breakage force on clamp catch trials of haptic task training. In contrast, in the haptic exposure experiment, pre-training with haptic task training only served to facilitate participants' learning in the subsequent haptic statistical exposure phase, in which they could also acquire an understanding of the rules of pulling, and so their haptic test performance itself was a reliable indicator of how much they understood these rules (as well as the identity of the pairs of the inventory). As we used the full range of haptic test performance to predict performance in the visual familiarity test (*Figure 2*, red lines, see also below), not understanding the rules of pulling could not lead to an erroneous negative finding. Therefore there was no need to exclude any of the participants in this experiment based on their performance on haptic task training. Nevertheless, we repeated all analyses by excluding participants based on the same criteria as in the other experiment (leading to the exclusion of only one participant), and all our results remained essentially unchanged, with small numerical modifications to the test statistics (not shown).

## Acknowledgements

We thank Daniel McNamee for technical assistance and Gergely Csibra and Richard Aslin for comments on the manuscript. This work was supported by an EU-FP7 Marie Curie CIG (JF), NIH R21 (JF), an ERC Consolidator Grant (ML), the Wellcome Trust (ML, DMW), and the Royal Society Noreen Murray Professorship in Neurobiology (DMW).

## Additional information

### Funding

| Funder | Grant reference number | Author |
|---|---|---|
| European Research Council | Consolidator Grant ERC-2016-COG/726090 | Máté Lengyel |

| Royal Society | Noreen Murray Professorship in Neurobiology RP120142 | Daniel M Wolpert |
| --- | --- | --- |
| Seventh Framework Programme | Marie Curie CIG 618918 | József Fiser |
| Wellcome Trust | New Investigator Award 095621/Z/11/Z | Máté Lengyel |
| National Institutes of Health | NIH R21 HD088731 | József Fiser |
| Wellcome Trust | Senior Investigator Award 097803/Z/11/Z | Daniel M Wolpert |

The funders had no role in study design, data collection and interpretation, or the decision to submit the work for publication.

## Author contributions

Gábor Lengyel, Conceptualization, Software, Formal analysis, Validation, Investigation, Visualization, Methodology, Writing—review and editing; Goda Žalalytė, Alexandros Pantelides, Data curation, Software, Formal analysis, Validation, Investigation, Visualization, Project administration; James N Ingram, Conceptualization, Software, Supervision, Investigation, Methodology, Project administration; József Fiser, Conceptualization, Resources, Supervision, Methodology, Writing—original draft, Writing—review and editing; Máté Lengyel, Conceptualization, Software, Formal analysis, Supervision, Funding acquisition, Validation, Investigation, Visualization, Methodology, Writing—original draft, Writing—review and editing; Daniel M Wolpert, Conceptualization, Resources, Software, Formal analysis, Supervision, Funding acquisition, Validation, Investigation, Visualization, Methodology, Writing—original draft, Project administration, Writing—review and editing

## Author ORCIDs

Gábor Lengyel https://orcid.org/0000-0002-1535-3250
Goda Žalalytė http://orcid.org/0000-0002-0012-9950
Alexandros Pantelides http://orcid.org/0000-0002-6234-6061
James N Ingram http://orcid.org/0000-0003-2567-504X
József Fiser https://orcid.org/0000-0002-7064-0690
Máté Lengyel https://orcid.org/0000-0001-7266-0049
Daniel M Wolpert http://orcid.org/0000-0003-2011-2790

## Ethics

Human subjects: All participants gave informed consent. All experimental protocols were approved by the University of Cambridge Psychology Ethics Committee.

## Decision letter and Author response

Decision letter https://doi.org/10.7554/eLife.43942.012
Author response https://doi.org/10.7554/eLife.43942.013

# Additional files

## Supplementary files

• Transparent reporting form
DOI: https://doi.org/10.7554/eLife.43942.008

## Data availability

The scripts for all the analyses reported in the manuscript can be found here https://github.com/GaborLengyel/Visual-Haptic-Statistical-Learning (copy archived at https://github.com/elifesciences-publications/Visual-Haptic-Statistical-Learning). There is a README file that explains both where the data can be found (Open Science Framework https://osf.io/456qb/) and how to run the analyses.

The following dataset was generated:

| Author(s) | Year | Dataset title | Dataset URL | Database and Identifier |
|---|---|---|---|---|
| Gábor Lengyel | 2019 | Visual-Haptic-Statistical-Learning | https://osf.io/456qb/ | Open Science Framework, 456qb |

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
