## [Decision Letter]

Thank you for submitting your article "Unimodal statistical learning produces multimodal object-like representations" for consideration by *eLife*. Your article has been reviewed by three peer reviewers including the Reviewing Editor, and the evaluation has been overseen by Michael Frank as the Senior Editor. The following individuals involved in review of your submission have agreed to reveal their identity: Michael Landy (Reviewer #1); Samuel J Gershman (Reviewer #2).

The reviewers have discussed the reviews with one another and the Reviewing Editor has drafted this decision to help you prepare a revised submission.

The paper "Unimodal statistical learning produces multimodal object-like representations" presents an elegant series of two experiments, demonstrating that human adults can make cross-modal inferences without bimodal training. Participants learned either haptically or visually that two particular shapes tend to form an object. In the visual condition the critical cue was the probability of the co-occurrence of the two shapes, in the haptic condition the force necessary to pull them apart. The authors show that these learned regularities generalize to a test in the other domain. Overall, we found the paper clearly written, interesting, and insightful.

1) While all reviewers agreed that the experiments provided new interesting insights in the learning and cross-modal generalization of learning, the theoretical link to the formation of objects is not as well motivated and may require a more careful treatment (see major comment from Sam Gershman and Jörn Diedrichsen). Specifically, the extant literature on the principles of object perception clearly provides evidence that boundary cues are important in the formation of objects (Spelke, 1990, etc.). While the authors show that statistical regularities can be learned and generalized across modalities, the paper does not provide any insight into the importance of boundary cues vs. statistical regularities. This should be better reflected in paper.

2) Yildirim and Jacobs, 2012, 2013, 2015; see also Erdogan, Yildirim and Jacobs, 2015, have developed computational models and provided empirical evidence for modality-independent representations of objects. This literature should be cited and the current work placed better in this context.

3) The reviewers and the reviewing editor were not 100% convinced by the argument that conscious awareness did not drive the learning/generalization effects. The authors do not even report the average score on the explicit test. What would help is to present data on explicit knowledge and generalization as a scatter plot. If all subjects showed high levels of performance on the explicit task, the validity of the partial correlation approach would be seriously called into question.

I am appending the original 3 reviews below for more details.

*Reviewing Editor:*

The paper "Unimodal statistical learning produces multimodal object-like representations" presents two elegant experiments, in which participants either haptically or visually learn that two particular shapes tend to form an object. In the visual condition the critical cue was the probability of the co-occurrence of the two shapes, in the haptic condition the force necessary to pull them apart. The authors show that these learned regularities then generalized to a test in the other domain.

Overall, I found the paper clearly written, interesting and insightful. I believe there are overall some weaknesses in the presentation of the results, and particularly in the strength of some of the statistical results (i.e. there is really no real evidence for a haptic -> visual generalization at the group level).

Furthermore, the motivation for the study is based a lot on how objects are learned in childhood. This arises two critical questions: Do the authors believe that the learning rules uncovered from an experiment in adults has anything to do with how 4-month old start to understand the world. Secondly, "objectness" is probably learned to a large degree by visual or haptic (as in surface touch) boundaries. Showing that adults CAN learn statistical regularities does not show that this is indeed the driving mechanism behind the acquisition of objects. Indeed, one might argue that statistical co-occurrence is not enough. For example, I often wear pants, but that doesn't make them and me one object?

*Reviewer #1:*

OK, this isn't typical reviewer-speak: This paper is really fun! It's such a good example of true synergistic collaboration, since it so obviously combines the diverse threads of the authors' previous work. And the result is a cool paradigm and a convincing set of data. I really don't have much in the way of useful comments, which is not typical for me. (OK, my comments might not usually be useful, but at least there are usually a lot of comments; ^) And that you managed to work in one of your chimera namesakes… Really, these are very well-designed, careful experiments. I wonder if the "objecthood" led observers to perceive, fuzzy as they might have been, something in the way of illusory contours between objects, especially since the location of these contours was cued in the haptic pulling training in the second experiment.

Subsection “Visual statistical exposure”: Here I did my usual OCD check on the methods and couldn't come up with 444 as the number of distinct possible stimuli. By my count, there are 8 each configurations with 3 horizontal or 3 vertical pair placements, and assuming you place the pairs without replacement, that's 2x8x6 = 96 stimuli. And, there are 14 configurations each with either 2 horizontal and 1 vertical or vice versa, yielding 2x14x(3x2x3) = 504 stimuli, for a total of 600. I'm not sure where I went wrong or how you got 444.

Subsection “Visual familiarity test”: I also checked the arithmetic here. It works, but it assumes that the pseudopairs take a left-side pattern from one true pair and combine it with a right-side pattern from another true pair, and keep them on their former sides. That's a good choice for keeping the pseudopair statistics similar to the true pair statistics, but you don't state explicitly that that's what you did.

Subsection “Haptic statistical exposure”, last paragraph: There are fewer scenes to learn in this experiment than in the visual training experiment and they were visible for much more time (not restricted to 700 ms).

Supplementary Figure 3: I'm glad this figure was included, although I'd put it in the main text. It's worthwhile seeing how much the pulling forces were modulated relative to the correct values. You might even cite this as a learning "gain" proportion (about 20%, to my eye).

*Reviewer #2:*

This paper presents an elegant series of experiments of visual-haptic integration, demonstrating that human adults can make cross-modal inferences without bimodal training. The authors argue that these results support theories of object perception based on learned statistical regularities. The paper is well-written and thorough. My main concerns are about conceptual foundations and novelty.

The definition of an object as "a material thing that can be seen and touched" is too weak, because it does not adhere to the constraints evident in infant (and adult) object cognition (as an aside, it's rather odd to use a dictionary definition when so much ink has been spilled on this question in cognitive science). Spelke, 1990, defines physical objects as "persisting bodies with internal unity and stable boundaries" (p. 30). She presents evidence that infant object perception adheres to four constraints: cohesion, boundedness, rigidity, and no action at a distance. It's important to note that a material thing can have statistical regularities without satisfying any of these constraints. For example, consider the jet of steam spouting from a boiling teapot. It's a material thing with statistical regularities (following gas laws, thermodynamics, etc.); it can be seen and touched. But it is not cohesive, bounded, or rigid. The experiments of Kellman and Spelke, 1983, and many others, show that the same arrangement of visible surfaces can be perceived as an object or not, depending on whether it adheres to these constraints.

To continue this point, the conceptual issue here is that the authors consider objects to be bundles of statistical regularities that support inferences about hidden properties: "participants should be able to predict haptic properties of objects based on just visual statistics, without any prior haptic experience with them, and vice versa, predict visual properties based on haptic statistical exposure alone." No doubt observed statistical regularities do support such inferences (this is what Spelke refers to as the "empiricist" account, which she criticizes). But the key insight from Spelke and others is that object perception goes beyond observed statistical regularities. There are inductive constraints on objecthood that support stronger generalizations. This conceptual issue could be resolved if the authors step back from their strong claims about the nature of objecthood, and instead focus on more restricted claims about how statistical learning shapes multimodal generalization.

I was surprised that the authors don't cite the work of Yildirim and Jacobs, 2012, 2013, 2015; see also Erdogan, Yildirim and Jacobs, 2015, which seems highly relevant. In fact, I feel that these prior papers diminish the novelty of the present paper, since Yildirim and colleagues showed empirical evidence for modality-independent representations of objects, and moreover they developed sophisticated computational models of how these representations are acquired and used.

[Editors' note: further revisions were requested prior to acceptance, as described below.]

Thank you for resubmitting your work entitled "Unimodal statistical learning produces multimodal object-like representations" for further consideration at *eLife*. Your revised article has been favorably evaluated by a Reviewing Editor and Michael Frank as the Senior Editor.

The manuscript has been improved to address most of the remaining issues. However, before final acceptance, I would like to request a couple of clarifications on the explicit awareness task to be presented in the paper (Materials and methods, and Figure 4 legend):

You say that "Participants were asked whether they noticed that the shapes came in pairs and if so then the shapes were presented to them individually and they had to point to the shapes that they thought were part of the inventory."

Does this mean that some of the participants were not presented with the objects if they said they didn't know? If yes, how many participants? Was their performance labeled with 0?

The information in the caption seems to indicate that "explicitness" is the proportion of correctly identified pairs out of 6+4 =10 possible pairs. Why are the proportion correct not in {0, 0.1, 0.2.…, 1}?

I assume the test was conducted by placing 12 (or 8) objects on the table and let the participants pick out candidate pairs (although the Materials and methods state that you showed the objects individually). Please clarify. Also, please indicate whether you asked the participants to "guess" remaining pairs or did you let them stop when they did not feel they knew?

If you forced participants to guess, please indicate chance performance on your explicit memory test.

If the participants were not forced to guess remaining pairs, please state this clearly. Given the standards in the memory literature, a force-choice test is the most stringent way to assess the presence/absence of explicit memory. Subjects may say they do not "know", but still be able to "guess" very much above chance. Without a forced-choice assessment, one is testing mostly meta-memory, but not explicit memory. So if this is the approach you took, I think the limitations of the explicit test should be pointed out in the Discussion.

---

## [Author Response]

[…] 1) While all reviewers agreed that the experiments provided new interesting insights in the learning and cross-modal generalization of learning, the theoretical link to the formation of objects is not as well motivated and may require a more careful treatment (see major comment from Sam Gershman and Jörn Diedrichsen). Specifically, the extant literature on the principles of object perception clearly provides evidence that boundary cues are important in the formation of objects (Spelke, 1990, etc.). While the authors show that statistical regularities can be learned and generalized across modalities, the paper does not provide any insight into the importance of boundary cues vs. statistical regularities. This should be better reflected in paper.

Please see our first response to the Reviewing Editor, and our first and third responses to reviewer #2 below.

2) Yildirim and Jacobs, 2012, 2013, 2015; see also Erdogan, Yildirim and Jacobs, 2015, have developed computational models and provided empirical evidence for modality-independent representations of objects. This literature should be cited and the current work placed better in this context.

Please see our fourth response to reviewer #2 below.

3) The reviewers and the reviewing editor were not 100% convinced by the argument that conscious awareness did not drive the learning/generalization effects. The authors do not even report the average score on the explicit test. What would help is to present data on explicit knowledge and generalization as a scatter plot. If all subjects showed high levels of performance on the explicit task, the validity of the partial correlation approach would be seriously called into question.

We have now added new analyses and include an additional Figure 4 which has all this information (and more). In particular, we show the distribution of explicitness across our participants (those who were debriefed) and that more than half had <50% explicitness, and also the details of the partial correlation analysis demonstrating that even after controlling for the effects of explicitness there is still a strong and highly significant correlation between performance on the two tasks in each experiment (although the baseline performance is different due to the unavoidable conflict between visual and haptic information in the haptic statistical exposure experiment).

I am appending the original 3 reviews below for more details comments.Reviewing Editor:[…] Overall, I found the paper clearly written, interesting and insightful. I believe there are overall some weaknesses in the presentation of the results, and particularly in the strength of some of the statistical results (i.e. there is really no real evidence for a haptic -> visual generalization at the group level).Furthermore, the motivation for the study is based a lot on how objects are learned in childhood. This arises two critical questions: Do the authors believe that the learning rules uncovered from an experiment in adults has anything to do with how 4-month old start to understand the world.

As we see it, there are really two questions bundled up in one here.

The first question is whether we believe that the statistical learning mechanisms demonstrated in our adult participants might also be present in infants. For this, our answer is that yes, we have good reasons to believe that is the case. To be clear, our statistical learning paradigm was designed to investigate sensory structure learning based purely on new statistics of the input. Importantly, we eliminated all typical low-level visual cues including Gestalt structures that could reveal the structure of the sensory input (and therefore help object formation). Exactly the same type of (albeit unimodal) statistical learning experiments were conducted in a large number of infant labs around the world with 5-8-month-old infants (Fiser and Aslin, 2002, Quinn and Bhatt, 2005, Saffran and Kirkham, 2018), and they found infants automatically and implicitly learn co-occurrence, conditional and chunk statistics. If we prefer parsimonious explanations, then assuming a close link between how adult and infants acquire new visual information with unfamiliar statistical constructs is a more feasible assumption than assuming two different mechanisms.

The second question is whether we believe these statistical learning mechanisms are actually at play in infants during development when they learn about new objects. For this, our answer is that we do not know, but we think there may be many real-life situations in which infants may need to learn about objects from ambiguous boundary information. In almost any natural scene there are many potential “boundaries” (edges, contrast transitions, etc) that are inside objects and thus do not correspond to actual boundaries separating objects (for example, think of a zebra: lots of boundaries but still one object). Conversely, objects may be separated by “illusory contours” (e.g. as in the Kanizsa triangle) which correspond to no physical boundary in the image but which we infer based on general knowledge about statistical regularities in our visual environment (Geisler et al., 2001). Thus, similar to pauses (auditory boundaries) in speech that have been argued to be poor predictors of where actual word boundaries are (Lehiste, 1970; Cole, 1980), edges in images will be unreliable predictors of where object boundaries are. Therefore, just as in the case of language acquisition, it is reasonable to assume that the statistical learning mechanisms we reveal will be also important for object learning in infants during development.

We have added this information into the Introduction (first paragraph) and Discussion (third paragraph).

Nevertheless, we agree that motivating our work from infant studies could have been misleading and we thus restructured the paper such that the Introduction focuses on the general importance and relevance of statistical learning mechanisms for segmenting the environment into objects, and we highlight potential links to the way infants identify and perceive objects only in the Discussion now (with a detailed discussion of the relationship between our work and the classical work of Spelke et al., see also our first response to reviewer #2).

Secondly, "objectness" is probably learned to a large degree by visual or haptic (as in surface touch) boundaries. Showing that adults CAN learn statistical regularities does not show that this is indeed the driving mechanism behind the acquisition of objects.

We have made it clear (Discussion, third paragraph) that our results show that statistics alone is sufficient to learn object-like representation. We feel that having shown this it will be for future studies to examine the relative role of statistics vs. other cues as a driving mechanism of the acquisition of objects.

Indeed, one might argue that statistical co-occurrence is not enough. For example, I often wear pants, but that doesn't make them and me one object?

As for the visual statistical exposure experiment, this is a misunderstanding of “co-occurrence” which we have now clarified in the paper (subsection “Visual stimuli”, last paragraph). Briefly, it’s not simply the number of times two things occur together but also the number of times they occur without each other that matters (see e.g. our earlier work for a formal definition, Orbán et al., 2008). For example, as opposed to the constituent shapes of a true pair in the experiment, which *always* appear together and *never* in a way that only one of them would be present, my pants sometimes appear without me (on the drying line, or the wardrobe), and I sometimes appear without that particular pair of pants (because I am wearing something else). In contrast, the legs of a pair of pants are part of one object, because they always appear together. As for the haptic exposure experiment, this example falls exactly into the class of pseudo-pairs as used in our experiment — things which have visual statistical co-occurrence but can be physically be separated easily so that they are not object-like.

Reviewer #1:[…] I wonder if the "objecthood" led observers to perceive, fuzzy as they might have been, something in the way of illusory contours between objects, especially since the location of these contours was cued in the haptic pulling training in the second experiment.

Interesting, but we did not explicitly test this.

Subsection “Visual statistical exposure”: Here I did my usual OCD check on the methods and couldn't come up with 444 as the number of distinct possible stimuli. By my count, there are 8 each configurations with 3 horizontal or 3 vertical pair placements, and assuming you place the pairs without replacement, that's 2x8x6 = 96 stimuli. And, there are 14 configurations each with either 2 horizontal and 1 vertical or vice versa, yielding 2x14x(3x2x3) = 504 stimuli, for a total of 600. I'm not sure where I went wrong or how you got 444.

We have included in the Materials and methods (subsection “Visual statistical exposure”) that the central square needed to be occupied by a shape so that snakes around the edge were not allowed (to minimize explicit cues) and location did not matter so that the same configuration of shapes translated were considered the same. This gives (where H and V are horizontal and vertical pairs respectively):

- 3H pairs gives

3 (identity of H1) ✕ 2 (identity of H2) ✕ 1(identity of H3) ✕ [1 (all aligned) + 2 (L/R displacement of one H) ✕ 3 (which H is displaced)] = 42

- 2H and 1V with the two H aligned gives

3 (identity of H1) ✕ 2 (identity of H2) ✕ 3 (identity of V) ✕ 2 (location of V on left/right) ✕ [1 (V aligned) + 2 (V offset top/bottom)]= 108

**-** 2H and 1V with the two H offset gives

3 (identity of H1) ✕ 2 (identity of H2) ✕ 2 (location of H1 offset L/R) ✕ 3 (identity of V) ✕ 2

(location of V above/below) = 72

Total = 222

And the same for 3V and 2V and 1H giving a total of 444 scenes.

We also added in to Figure 1—figure supplement 1 all 444 configurations used as we thought that would be useful.

Subsection “Visual familiarity test”: I also checked the arithmetic here. It works, but it assumes that the pseudopairs take a left-side pattern from one true pair and combine it with a right-side pattern from another true pair, and keep them on their former sides. That's a good choice for keeping the pseudopair statistics similar to the true pair statistics, but you don't state explicitly that that's what you did.

We have added this to the Materials and methods (subsection “Visual familiarity test”). However, the comment “keep them on their former sides” is not correct as shapes of horizontal true pairs are used to construct vertical pseudo pairs, and vice versa.

Subsection “Haptic statistical exposure”, last paragraph: There are fewer scenes to learn in this experiment than in the visual training experiment and they were visible for much more time (not restricted to 700 ms).

We have added a sentence to this effect (subsection “Haptic statistical exposure”, last paragraph).

Supplementary Figure 3: I'm glad this figure was included, although I'd put it in the main text. It's worthwhile seeing how much the pulling forces were modulated relative to the correct values. You might even cite this as a learning "gain" proportion (about 20%, to my eye).

We have included the difference in average force in the legend of Figure 3.

Reviewer #2:[…] The definition of an object as "a material thing that can be seen and touched" is too weak, because it does not adhere to the constraints evident in infant (and adult) object cognition (as an aside, it's rather odd to use a dictionary definition when so much ink has been spilled on this question in cognitive science). Spelke, 1990, defines physical objects as "persisting bodies with internal unity and stable boundaries" (p. 30). She presents evidence that infant object perception adheres to four constraints: cohesion, boundedness, rigidity, and no action at a distance. It's important to note that a material thing can have statistical regularities without satisfying any of these constraints. For example, consider the jet of steam spouting from a boiling teapot. It's a material thing with statistical regularities (following gas laws, thermodynamics, etc.); it can be seen and touched. But it is not cohesive, bounded, or rigid. The experiments of Kellman and Spelke, 1983, and many others, show that the same arrangement of visible surfaces can be perceived as an object or not, depending on whether it adheres to these constraints.

While we had already cited the work of Spelke et al. in the previous version of the manuscript (including both references the reviewer mentions), we agree that we could have better clarified how we see the relationship between our work and theirs. Briefly, we see our work as complementary to that of Spelke et al.: we are interested in showing that humans (including, presumably, infants, see our first response to the Reviewing Editor) can use primarily statistical cues to segment the world into constituent components which pass a fundamental *necessary* criterion for objects – that of across-modality generalization (which, btw, we see as equivalent to what the reviewer cites as Spelke’s requirement for “going beyond observed statistical regularities”, see below). In contrast, Spelke et al. defined a specific set of criteria that infants use to identify objects. If, as we argue (see our first response to the Reviewing Editor), the statistical learning mechanisms we revealed also operate in infants then our results would suggest that Spelke’s criteria may be *sufficient* but not *necessary*.

For example, one might argue that the objects in our experiment violated even the basic requirement of having 3-dimensional structure, and specifically Spelke’s principle of “cohesion” because their constituent shapes were separated by gaps (although in front of a spatially contiguous gray background). Thus, Spelke’s criteria may be special cases of a more general principle of statistical coherence. Importantly, previous work on statistical learning only showed that statistical cues can be used for segmentation (in various sensory modalities), but did not investigate the “objectness” of the resulting representations in any way. Therefore, we think our work represents an important first step in connecting the field of statistical learning to the kind of object representations that Spelke (and others) have studied. It may also well be that an internal representation segmented based on statistical coherence (and other cues) needs to eventually pass a number of additional criteria (e.g. those that Spelke et al. have identified) to become a real mental object, and it will be for future studies to test whether and how statistical learning mechanisms can produce such representations. As for the jet of steam example: we think that there are many shades of perceiving “objectness” (ranging from rigid bodies through more elusive entities, such as a melting scoop of ice cream or the reviewer’s example of a jet of steam of a boiling teapot, to collections of clearly separate objects), and further work will be needed to refine the necessary and sufficient conditions for segmenting entities with different degrees of objectness on this continuum. Note that we have also been careful to call the representations we demonstrate “object-like” rather than “object”, and we now also discuss these issues more carefully and in more detail in the Discussion (third paragraph).

We also agree that motivating our work from infant studies could have been misleading and we thus restructured the paper such that the Introduction focuses on the general importance and relevance of statistical learning mechanisms for segmenting the environment into objects, and we highlight potential links to the way infants identify and perceive objects only in the Discussion now (see also our first response to the Reviewing Editor).

To continue this point, the conceptual issue here is that the authors consider objects to be bundles of statistical regularities that support inferences about hidden properties: "participants should be able to predict haptic properties of objects based on just visual statistics, without any prior haptic experience with them, and vice versa, predict visual properties based on haptic statistical exposure alone." No doubt observed statistical regularities do support such inferences (this is what Spelke refers to as the "empiricist" account, which she criticizes). But the key insight from Spelke and others is that object perception goes beyond observed statistical regularities.

We agree. Importantly, this is precisely what we also show with zero-shot generalization: that even if only visual statistical regularities are observed, participants can infer haptic properties, and vice versa. We now make this clear in the text (Discussion, third paragraph).

There are inductive constraints on objecthood that support stronger generalizations. This conceptual issue could be resolved if the authors step back from their strong claims about the nature of objecthood, and instead focus on more restricted claims about how statistical learning shapes multimodal generalization.

We hope to have clarified our claims here in our response, and in the Introduction and Discussion of the manuscript which we have now substantially rewritten, and that these are now in agreement with the reviewer’s perspective.

I was surprised that the authors don't cite the work of Yildirim and Jacobs, 2012, 2013, 2015; see also Erdogan, Yildirim and Jacobs, 2015, which seems highly relevant. In fact, I feel that these prior papers diminish the novelty of the present paper, since Yildirim and colleagues showed empirical evidence for modality-independent representations of objects, and moreover they developed sophisticated computational models of how these representations are acquired and used.

Indeed, while we were aware of Jacobs et al.’s relevant work (Yildirim and Jacobs 2012, 2013, 2015, Erdogan, Yildirim and Jacobs 2015), we failed to include these in our reference list and have corrected this in the new manuscript (Discussion, second and last paragraphs). However, we respectfully disagree with the reviewer on how much this body of work diminishes the novelty of our results. While all four papers from the Jacobs lab focus on multi-modal sensory information processing, Yildirim and Jacobs, 2012 is a purely modeling paper, without any empirical data, based on computational principles we fully agree with, Yildirim and Jacobs, 2015 is about a special case of sequence learning, and Erdogan, Yildirim and Jacobs, 2015 has no learning, just similarity ranking. The most directly relevant to our work is Yildirim and Jacobs, 2013, however, even that work only concerned objects that were already fully segmented by clear boundaries and so it does not speak to the central issue we investigate here: the role of statistical learning mechanisms in segmenting the work into object-like entities.

[Editors' note: further revisions were requested prior to acceptance, as described below.]

The manuscript has been improved to address most of the remaining issues. However, before final acceptance, I would like to request a couple of clarifications on the explicit awareness task to be presented in the paper (Materials and methods, and Figure 4 legend):

We have now rewritten this part of the Materials and methods with substantially more detail to clarify the issues below (subsections “Debriefing” and ׅ“Controlling for explicit knowledge of pairs”).

You say that "Participants were asked whether they noticed that the shapes came in pairs and if so then the shapes were presented to them individually and they had to point to the shapes that they thought were part of the inventory." Does this mean that some of the participants were not presented with the objects if they said they didn't know? If yes, how many participants? Was their performance labeled with 0?

If participants said they did not know the shapes came in pairs (8 out of 23 participants) we did not present them with the shapes and they scored 0 in our explicitness measure. All participants who reported that shapes came in pairs were able to identify at least one pair correctly.

The information in the caption seems to indicate that "explicitness" is the proportion of correctly identified pairs out of 6+4 =10 possible pairs. Why are the proportion correct not in {0, 0.1, 0.2.…, 1}?

Different participants were used in the two experiments with 6 pairs in the visual exposure and 4 true pairs in the haptic exposure experiment so that the scores were multiples of 1/6 and 1/4 for the two experiments. These are combined in the plot.

I assume the test was conducted by placing 12 (or 8) objects on the table and let the participants pick out candidate pairs (although the Materials and methods state that you showed the objects individually). Please clarify.

That is correct – if by “objects” you mean shapes and by “individually” you mean not attached to each other. See also expanded explanation in the text (subsection “Debriefing”).

Also, please indicate whether you asked the participants to "guess" remaining pairs or did you let them stop when they did not feel they knew?If you forced participants to guess, please indicate chance performance on your explicit memory test.

We did not ask them to guess the remaining pairs and they could stop when they had indicated all pairs they could identify. The score was only based on correctly identified pairs and we ignored incorrect pairs so as to err on the side of increased explicitness in our measure.

If the participants were not forced to guess remaining pairs, please state this clearly. Given the standards in the memory literature, a force-choice test is the most stringent way to assess the presence/absence of explicit memory. Subjects may say they do not "know", but still be able to "guess" very much above chance. Without a forced-choice assessment, one is testing mostly meta-memory, but not explicit memory. So if this is the approach you took, I think the limitations of the explicit test should be pointed out in the Discussion.

We have now explicitly(!) clarified in all corresponding parts of the text that our measure of explicitness measures *the sense of knowing* that there are pairs as well as being able to identify them (in the last paragraph of the Results, Figure 4 legend, the first paragraph of the Discussion, and the sections on Debriefing and Data analysis in the Materials and methods, subsections “Debriefing” and ׅ“Controlling for explicit knowledge of pairs”). To some extent the suggestion to perform forced choice is similar to what happened in the familiarity part of the experiment where we presented a true pair and a mixture pair and asked which was more familiar. We did this for all true pairs. We believe a key difference is that in the familiarity test we did not explicitly tell participants that there were true pairs in the experiment at all. This measure of familiarity is commonly accepted to test *implicit* learning. The debriefing was really to rule out that highly cognitive operations accounted for all the learning.

Finally, in preparing the data to be shared online we found a small coding error in the stimulus generation script and have added the following text to the Materials and methods:

“Although the assignment of shapes to objects (pairs) was randomized across participants, we found at the end of the experiments that some participants had the same order of trials due to a coding error. […] There is no reason to believe that the order of trials would affect learning or performance.”